# GeoFlow: Geo-Aware Modeling of Inter-Area Relationships in OD Flow Prediction and Generation

## Abstract

Origin–destination (OD) flow modeling underpins urban planning and mobility analysis, but prevailing graph-based methods often neglect salient geographic attributes, limiting their ability to model long-range and multi-area dependencies. In this paper, we introduce GeoFlow, a novel framework that (i) augments area representations with geospatial attributes, including relative positions, $k$-hop and geodesic distances, (ii) employs a specialized geometric-intrinsic fusion encoder design that combines graph attention for intrinsic area signals with coordinate-aware encoders for global structure, and (iii) adopts an axial-global attention decoder to capture OD-specific competitive dependencies. For OD flow generation, GeoFlow is paired with flow matching models to produce more authentic and diverse mobility samples. Empirically, GeoFlow achieves superior performance in predictive accuracy, while substantially improving generative fidelity and diversity. Ablation and analytical studies confirm the contribution of each component. Code is open-source and available at this URL.

## 1 Introduction

Origin–destination (OD) flow describes population movements between areas, reflecting social and economic activity at a macro level and supporting applications such as urban planning and behavioral analysis (Batty, 2007; Zhang et al., 2021; Wu et al., 2024). However, large-scale data collection faces statistical and privacy challenges (Simini et al., 2021; Long et al., 2023), and generalizable models are needed to inform urban development, especially in data-scarce or early-planning regions (Rong et al., 2025). Consequently, OD flow prediction and generation have become central research topics (Luca et al., 2021). Prediction methods estimate flows from area attributes (e.g., socio-demographics), while generation methods further employ random seeds and generative models to capture diverse mobility patterns (Liu et al., 2020; Rong et al., 2024). Despite their conceptual differences, both tasks share common modeling principles, and methods are often evaluated jointly (Rong et al., 2023; 2025). Early approaches, including gravity (Zipf, 1946) and radiation models (Simini et al., 2012), were followed by machine learning techniques such as Random Forest (Breiman, 2001; Pourebrahim et al., 2019) and SVR (Drucker et al., 1996; Rodríguez-Rueda et al., 2021). With the rise of deep learning (Dong et al., 2021), models such as DGM (Simini et al., 2021) and GMEL (Liu et al., 2020) achieved notable progress, and recent graph-based methods further advanced prediction and generation by modeling areas as attributed nodes and OD flows as directed weighted edges (Bojchevski et al., 2018; Liu et al., 2020; Rong et al., 2025).

Although existing state-of-the-art methods represent area networks as graph structures to exploit properties such as translational and rotational equivalent (Liu et al., 2020; Satorras et al., 2021), this representation often overlooks easily accessible geographic information, thereby complicating the learning process (Klemmer et al., 2023). In practice, relationships between areas are encoded in matrix form through pairwise adjacency and straight-line distances (Luo et al., 2024; Rong et al., 2025). While this formulation is theoretically lossless and supports network reconstruction, it fails to directly preserve fundamental geometric properties such as collinearity, which can be readily obtained from coordinate systems, and geodesic distances, which are essential but difficult to infer from pairwise matrices (Whiteley et al., 2021). Recovering such properties is challenging, and their absence undermines the model's ability to capture long-range and multi-area dependencies that are

crucial for OD flow modeling (Alon & Yahav, 2020). Consequently, current feature representation and encoding strategies remain limited in representing global area-to-area relationships.

This limitation highlights a central challenge in OD flow modeling: how to represent and encode complex inter-area relationships. Graph structures offer desirable invariances and computational advantages, yet they often fail to capture long-range and multi-area geographic interactions (Bamberger et al., 2025). Coordinate-based representations encode geometric attributes directly but are sensitive to the choice of reference origin, scale, and orientation (Bronstein et al., 2021). Besides, some geospatial attributes (e.g., geodesic distance) remain nontrivial to recover from matrix representations (Whiteley et al., 2021). Accordingly, integrating complementary representation methods and developing encoding–decoding strategies that preserve geometric attributes while remaining insensitive to reference choices are essential for improving OD flow prediction and generation.

This paper introduces GeoFlow, a novel framework for OD flow prediction and generation that systematically integrates geospatial attributes and encoder–decoder design. We first perform geospatial attribute augmentation by incorporating $k$-hop distance, geodesic distance, and relative position alongside straight-line distance, allowing the model to access key spatial relationships directly rather than extracting them from matrices. In the encoder, we propose *geometry-intrinsic fusion* encoder, combining Cartesian coordinates with graph-based representations: a graph attention module encodes intrinsic area attributes (e.g., points of interest) and aggregates influences from neighboring areas, while carefully chosen coordinate centers and orientations preserve relative positions, enabling the model to capture long-range and multi-area dependencies. In the decoder, we introduce an *axial-global attention* mechanism that models competitive relationships between areas while reducing computational costs, making joint attention across hundreds of areas feasible. Finally, for flow generation, we adopt flow matching models, which provide higher stability, efficiency, and generalization than diffusion models (Lipman et al., 2022; Huang et al., 2025). To the best of our knowledge, this is the first work to apply flow matching in the OD flow generation task.

Extensive experiments demonstrate that GeoFlow markedly improves both OD flow prediction and generation. For the prediction task, it consistently achieves substantially lower errors than existing baselines, reflecting a clear advantage. In the generation task, GeoFlow attains a 7.4% relative improvement in reconstruction accuracy and produces more authentic and diverse samples. Ablation studies on geospatial representations show that even lightweight attribute augmentations provide notable gains, highlighting the practical value of these previously overlooked features, and ablation experiments on the encoder and decoder architectures confirm their effectiveness. Furthermore, analytical experiments demonstrate the rationale behind the overall design and offer deeper insight into the role of geospatial attributes.

Our contributions can be summarized in three main parts:

1. We propose GeoFlow, a novel method for OD flow prediction and generation that systematically enhances feature representation and model architecture. Each area is augmented with crucial geospatial attributes to capture long-range and multi-area relationships. A geometric-intrinsic fusion encoder module encodes area properties, while an axial-global attention mechanism models competitive dependencies among areas.

2. We evaluate GeoFlow on both OD flow prediction and generation tasks. GeoFlow achieves substantial improvements in prediction accuracy over baselines and a 7.4% relative gain in reconstruction accuracy for generation, producing more authentic and diverse samples.

3. We conduct ablation experiments on each model component, showing that the proposed approaches substantially improve model performance. Additionally, analytical experiments on the geospatial features provide deeper insights into the dynamics of OD flow modeling.

## 2 RELATED WORK

Existing work on origin–destination (OD) flow prediction or generation falls into two broad categories: principle-driven methods and data-driven methods. **Principle-driven methods** derive flows from explicit mechanistic physical models and statistical formulations. Zipf (1946) employs the gravity model to explain intercity human mobility. Tomazinis (1962) introduces the notion of competitive opportunities and incorporates probability theory to estimate OD flows. Simini et al. (2012)

propose a parameter-free radiation model and enhance the accuracy of flow modeling. These formulations provide transparent explanations of aggregate flows but lack flexibility for complex empirical patterns. **Data-driven methods**, which learn flow patterns directly from observed mobility data, have emerged and revitalized the field in recent years. Many studies apply classical machine learning algorithms to OD flow generation. Random Forest (RF) demonstrates strong potential in the task (Pourebrahim et al., 2019). Gradient Boosting Regression Trees (GBRT) enhance predictive accuracy through boosting and have been effectively applied to urban OD prediction (Robinson & Dilkina, 2018). Support Vector Regression (SVR) employs kernel methods to estimate flows between regions based on their urban attributes (Rodríguez-Rueda et al., 2021). Despite their demonstrated success, classical machine learning methods have limitations in fully capturing the complex dynamics and high-dimensional interactions inherent in urban OD flows, paving the way for deep learning-based techniques. The Deep Gravity Model (DGM), inspired by traditional gravity models, uses multi-layer perceptrons to estimate flow distributions (Simini et al., 2021). Graph-based methods, including NetGAN (Bojchevski et al., 2018) and Geographical Multi-task Embedding Learning (GMEL) (Liu et al., 2020), leverage random walks or graph neural networks to capture spatial and structural information, enabling more accurate OD flow generation. More recent approaches, such as DiffODGen (Rong et al., 2023) and WEDAN (Rong et al., 2025), employ diffusion models to model graph topology and edge weights conditioned on urban attributes. Despite their ability to capture complex spatial and structural patterns, these methods typically abstract areas as attributed nodes, overlooking spatial proximity and inter-area interactions and thereby hindering further advances. TransFlower (Luo et al., 2024) acknowledges the importance of relative location in capturing directional relations between areas; however, its reliance on pairwise attention and limited integration of topological structure restricts its capacity to model OD flows.

## 3 METHOD

### 3.1 PROBLEM DEFINITION AND FORMULATION

We consider OD flow modeling as a unified paradigm: the core problem is to characterize flows among areas under given attributes. The paradigm can be instantiated in two ways: (1) When the goal is to estimate missing or future entries, the formulation reduces to prediction; (2) When the goal is to capture the full stochastic behavior of flows, it becomes generation. Both tasks share the same underlying setting but differ in the learning objectives.

Formally, let $R$ denote a region partitioned into $N$ non-overlapping areas $\{A_i\}_{i=1}^N$; each area $A_i$ is associated with a feature vector $X_i \in \mathbb{R}^d$, and the collection is denoted by $\mathbf{X} = [X_1; \ldots; X_N] \in \mathbb{R}^{N \times d}$. OD flows are represented as a non-negative matrix $\mathbf{F} \in \mathbb{R}_{\geq 0}^{N \times N}$ where entry $F_{ij}$ specifies the flow from $A_i$ to $A_j$, while inter-area geographic or topological information is captured by $G$ (e.g., adjacency $\mathbf{W}$, distance matrix $\mathbf{D}$, or coordinates $\{p_i\}$).

**Prediction as supervised estimation.** Given a training index set $\Omega \subseteq \{1, \ldots, N\}^2$, the prediction task seeks to estimate $\hat{\mathbf{F}} = f_\theta(\mathbf{X}, G)$ by minimizing the discrepancy between observed and predicted flows:

$$\min_\theta \ \mathcal{L}_{\text{pred}}(\theta) = \frac{1}{|\Omega|} \sum_{(i,j) \in \Omega} \ell(F_{ij}, \hat{F}_{ij}) \quad \text{s.t. } \hat{F}_{ij} \geq 0 \ \forall i, j, \tag{1}$$

where $\ell(\cdot, \cdot)$ is a per-entry loss (e.g., RMSE).

**Generation as distributional modeling.** Beyond point estimation, the generation task requires capturing the full distributional structure of OD flows. This can be formulated via a conditional generator $g_\theta(z; \mathbf{X}, G)$, with latent seed $z \sim p_z$, optimized under

$$\min_\theta \ \mathcal{L}_{\text{gen}}(\theta) = \mathbb{E}_{\mathbf{F} \sim p_{\text{data}}} \mathbb{E}_{z \sim p_z} \left[ \ell_{\text{rec}}(\mathbf{F}, g_\theta(z; \mathbf{X}, G)) \right]$$
$$+ \alpha \, \mathcal{L}_{\text{dist}}(p_{\text{data}}, p_\theta) + \beta \, \mathcal{L}_{\text{div}}(\{g_\theta(z_k; \mathbf{X}, G)\}_{k=1}^K), \tag{2}$$

where $\ell_{\text{rec}}$ enforces fidelity (e.g., RMSE), $\mathcal{L}_{\text{dist}}$ measures distributional discrepancy (e.g., Jensen–Shannon divergence) between $p_{\text{data}}$ and $p_\theta$, and $\mathcal{L}_{\text{div}}$ promotes diversity. Hyperparameters $\alpha, \beta \geq 0$ balance reconstruction fidelity, distributional realism, and sample diversity.

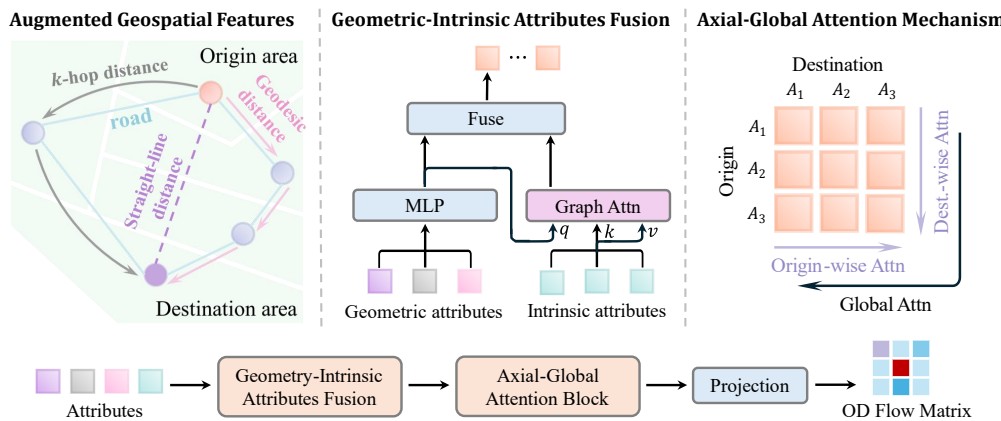

Figure 1: **Framework of GeoFlow.** The framework comprises three main components: the augmented geospatial relation descriptors (upper left), the encoder for fusing geometric and intrinsic attributes (upper middle), and the axial–global attention–based decoder (upper right).

## 3.2 GEOFLOW

Unlike trajectory forecasting, which requires reconstructing detailed travel paths, OD flow tasks focus on aggregate movements between areas. A straightforward baseline concatenates origin and destination features to predict or generate flows from pairwise representations. However, this formulation treats areas as isolated nodes and neglects critical geographic and multi-area dependencies.

To address these limitations, we partition area attributes into two categories and design dedicated encoders. The first, *geometric attributes*, includes coordinates, pairwise distances, and spatial descriptors such as geodesic proximity, which support global and long-range interaction modeling. The second, *intrinsic attributes*, covers local covariates such as points of interest (POIs) and socio-demographics, which characterize production and attraction capacity and primarily affect nearby areas. These two categories differ in semantics, scale, and propagation behavior: geometric attributes impose structural and long-range constraints, whereas intrinsic attributes capture localized demand and short-range interactions. Accordingly, we encode them separately and fuse their representations downstream to integrate global structure with local context. For decoding, we introduce an axial attention mechanism that models competitive dependencies in human mobility: OD pairs sharing the same origin or destination exhibit stronger correlations than unrelated pairs. This mechanism is further combined with global attention, resulting in a decoder that enhances training efficiency while achieving superior performance within comparable computational budgets. Fig. 1 illustrates the overall architecture of the GeoFlow framework.

The following sections elaborate on these components: Sec. 3.2.1 describes the additional geospatial attributes considered for OD flow tasks, Sec. 3.2.2 presents the encoding strategies for the two categories, and Sec. 3.2.3 details the axial-global attention mechanism.

### 3.2.1 GEOSPATIAL FEATURE AUGMENTATION

Since graph-based approaches typically store geographical relationships as adjacency or distance matrices, recovering relative positions or geodesic distances from such representations is difficult, limiting the model's ability to capture essential factors that shape travel behavior (Whiteley et al., 2021; Klemmer et al., 2023). Rather than relying solely on deeper or wider networks, explicitly providing these salient features offers a direct and cost-effective alternative, while also supplying inductive signals that stabilize and accelerate training.

In this work, we enrich each area with three additional attributes: **relative position**, **$k$-hop distance**, and **geodesic distance**. The relative position is defined as the coordinate difference between two areas under a Cartesian system assigned to the region. To ensure translational and rotational invariance while retaining meaningful orientation, we place the origin at the region's geometric center, align the $x$-axis with the principal orientation (the direction of greatest variance), and normalize coordi-

nates to $[-1, 1]$, thereby mitigating scale differences across regions. The $k$-hop distance is the length of the shortest unweighted path between two areas, reflecting topological proximity. The geodesic distance, or free-flow distance, is the minimal travel length along the network, capturing realistic accessibility. Fig. 1 upper left illustrates an example of the augmented geospatial features. By incorporating these attributes, GeoFlow provides richer geographic context, improving both predictive accuracy and training stability in OD flow modeling.

### 3.2.2 ENCODER: GEOMETRIC–INTRINSIC FUSION

To effectively leverage the enriched geospatial features, GeoFlow employs a dedicated encoder that fuses geometric and intrinsic area attributes. As shown in Fig. 1 upper middle, for the geometric attributes, which include relative position, straight-line distance, $k$-hop distance, and geodesic distance, we encode each pairwise attribute vector using a multilayer perceptron (MLP). These geometric attributes are normalized to remove scale-induced biases. By adopting a consistent regional coordinate system and applying systematic normalization, the model can more readily infer derived geometric cues (e.g., relative bearing) within a fixed domain. Normalized relative positions also reduce biases introduced by absolute location and orientation, obviating the need for explicit trigonometric encodings. For a region with $N$ areas, let the geometric information be denoted by the tensor $G \in \mathbb{R}^{N \times N \times d_g}$, where $d_g$ is the dimension of the per-pair geographic descriptor. GeoFlow employs an MLP $\phi_g : \mathbb{R}^{d_g} \to \mathbb{R}^{d_e}$ elementwise to obtain an encoded geographic embedding

$$E_g = \phi_g(G) \in \mathbb{R}^{N \times N \times d_e}, \quad E_g[i,j,:] = \phi_g\big(G[i,j,:]\big). \tag{3}$$

For intrinsic attributes, given their dependence on neighboring areas, we adopt a graph-attention encoding (Veličković et al., 2017). Unlike prior approaches that rely solely on local message passing and are therefore constrained by network depth, GeoFlow introduces an axial attention design to enhance information propagation efficiency. An area's influence on others depends on both its intrinsic attributes and geographic relations; accordingly, GeoFlow conditions attention on the geographic tensor when computing inter-area interactions. Concretely, let $X \in \mathbb{R}^{N \times d_x}$ denote the intrinsic attribute matrix. GeoFlow first projects $X$ as $E_x$ and then updates $E_x$ via an axial attention operator,

$$\widetilde{E_x}[i,:] = \sum_{k=1}^{N} \alpha_{ij} E_x[j,:], \quad \alpha_{ij} = \mathrm{softmax}\left([\mathcal{A}_{\mathrm{global}}^{\mathrm{enc}} \circ \mathcal{A}_{\mathrm{axial}}^{\mathrm{enc}}](S)\right)[i,j],$$

$$S[i,j] = w_{\mathrm{geo}}^{\top} G[i,j,:], \quad E_x = X W_x, \tag{4}$$

where $\mathcal{A}_{\mathrm{axial}}^{\mathrm{enc}}$ denotes the combined origin-wise and destination-wise attention and $\mathcal{A}_{\mathrm{global}}^{\mathrm{enc}}$ denotes the global attention mechanism that computes attention weights between areas conditioned on the geographic tensor $G$. $w_{\mathrm{geo}} \in \mathbb{R}^{d_g}$ is a learnable vector that compresses the per-pair geographic descriptor, turning $G$ into a score matrix $S \in \mathbb{R}^{N \times N}$, and $W_x \in \mathbb{R}^{d_x \times d_x}$ is a learnable projection matrix. The encoder-side axial–global attention mechanism (i.e., $\mathcal{A}_{\mathrm{axial}}^{\mathrm{enc}}$ and $\mathcal{A}_{\mathrm{global}}^{\mathrm{enc}}$) shares the same form as its decoder counterpart, which we describe in detail later in Sec. 3.2.3, but in the encoder it operates on the geographic tensor $G$ as input and is parameterized independently.

Finally, we employ an MLP to integrate the geographic embedding $E_g$ and the intrinsic embedding $\widetilde{E_x}$, producing the joint representation $Z$. Formally, let $\phi_f : \mathbb{R}^{d_e + d_x} \to \mathbb{R}^{d_z}$ denote the fusion MLP. For each OD pair $(i, j)$, the fused embedding is obtained as

$$Z[i,j,:] = \phi_f\Big( [E_g[i,j,:] \parallel \widetilde{E_x}[i,:] \parallel \widetilde{E_x}[j,:]] \Big), \tag{5}$$

where $\parallel$ denotes concatenation. The resulting tensor $Z \in \mathbb{R}^{N \times N \times d_z}$ serves as the unified representation for subsequent OD flow prediction and generation.

### 3.2.3 DECODER: AXIAL–GLOBAL ATTENTION

Naive full attention can capture interactions among OD pairs (Vaswani et al., 2017). Still, it is computationally prohibitive: for a region with $N$ areas, the $N^2$ OD pairs induce an $O(N^4)$ cost for pairwise attention, which limits the depth of dynamics the model can afford under practical budgets (Ho et al., 2019). Empirically, OD pairs that share an origin or share a destination are more

strongly coupled. Fixing an origin induces comparisons among candidate destinations, while a destination's finite capacity differentiates travelers from different origins, underscoring the relevance of origin- and destination-conditioned interactions. Accordingly, we adopt an axial attention scheme that alternately applies origin-wise and destination-wise attention for stacked blocks. This design efficiently captures origin-conditioned and destination-conditioned relational patterns without evaluating full pairwise attention, reducing computational cost to $O(N^3)$. The decoder takes the unified OD representation $Z$ from Eq. 5 as input and refines it with a stack of $L_a$ axial-attention layers interleaved with global-attention layers. Each axial layer consists of an origin-wise sublayer followed by a destination-wise sublayer, and thus performs two successive updates of the OD tensor. We initialize $Z^{(0)} = Z$ and, for $k = 0, 1, \ldots, L_a - 1$, use $Z^{(2k)}$ and $Z^{(2k+1)}$ to denote the inputs to the $(k+1)$-th origin-wise and destination-wise attention sublayers, respectively. For notational convenience, we write $Z_{i,:}^{(m)} = Z^{(m)}[i, :, :] \in \mathbb{R}^{N \times d_z}$ for the slice corresponding to origin $i$ and $Z_{:,j}^{(m)} = Z^{(m)}[:, j, :] \in \mathbb{R}^{N \times d_z}$ for the slice corresponding to destination $j$.

We then define the axial-attention mechanism for the $(k+1)$-th layer as

$$\mathcal{A}_{\text{origin}}^{(k+1)}(Z^{(2k)})[i, :, :] = \text{softmax}\left( \frac{(Z_{i,:}^{(2k)} W_{o_q}^{(k+1)})(Z_{i,:}^{(2k)} W_{o_k}^{(k+1)})^\top}{\sqrt{d_z}} \right) (Z_{i,:}^{(2k)} W_{o_v}^{(k+1)}), \quad (6)$$

$$\mathcal{A}_{\text{dest}}^{(k+1)}(Z^{(2k+1)})[:, j, :] = \text{softmax}\left( \frac{(Z_{:,j}^{(2k+1)} W_{d_q}^{(k+1)})(Z_{:,j}^{(2k+1)} W_{d_k}^{(k+1)})^\top}{\sqrt{d_z}} \right) (Z_{:,j}^{(2k+1)} W_{d_v}^{(k+1)}), \quad (7)$$

where $W_{o_q}^{(k+1)}, W_{o_k}^{(k+1)}, W_{o_v}^{(k+1)}, W_{d_q}^{(k+1)}, W_{d_k}^{(k+1)}, W_{d_v}^{(k+1)} \in \mathbb{R}^{d_z \times d_z}$ are learnable projection matrices for queries, keys and values in the $(k+1)$-th layer. This indexing yields $2L_a$ intermediate tensors $Z^{(0)}, \ldots, Z^{(2L_a)}$ by successively setting $Z^{(2k+1)} = \mathcal{A}_{\text{origin}}^{(k+1)}(Z^{(2k)})$ and $Z^{(2k+2)} = \mathcal{A}_{\text{dest}}^{(k+1)}(Z^{(2k+1)})$, with $Z^{(2L_a)}$ being the output of the axial stack. We denote the $L_a$-block axial attention by the operator

$$\mathcal{A}_{\text{axial}}^{L_a}(Z) = \left( \prod_{k=0}^{L_a-1} \left[ \mathcal{A}_{\text{dest}}^{(k+1)} \circ \mathcal{A}_{\text{origin}}^{(k+1)} \right] \right)(Z), \quad (8)$$

so that $Z^{(2L_a)} = \mathcal{A}_{\text{axial}}^{L_a}(Z)$. Although $\mathcal{A}_{\text{axial}}$ is defined here on the OD embedding tensor $Z$, it is an operator over generic three-dimensional tensors (and, via a dimension unsqueeze operation, also applicable to two-dimensional inputs). We use the same axial formulation on the encoder side when modeling geographic relations in Eq. 4.

To avoid losing the complex, multi-area interactions among OD pairs, we also introduce a computationally and memory-efficient global-attention pathway that complements the axial blocks. Instead of applying full attention over all $N^2$ OD pairs, we first aggregate OD information at the area level. Given $Z \in \mathbb{R}^{N \times N \times d_z}$, we compute origin-wise and destination-wise pooled representations and concatenate them as

$$H^{\text{orig}}[i, :] = \frac{1}{N} \sum_{j=1}^{N} Z[i, j, :], \quad H^{\text{dest}}[j, :] = \frac{1}{N} \sum_{i=1}^{N} Z[i, j, :], \quad H = \phi_g\left( [H^{\text{orig}} \| H^{\text{dest}}] \right), \quad (9)$$

where $H^{\text{orig}}, H^{\text{dest}} \in \mathbb{R}^{N \times d_z}$ summarize OD information for each origin and destination, respectively, and $\phi_g : \mathbb{R}^{2d_z} \to \mathbb{R}^{d_z}$ is a projection MLP applied row-wise, so that $H \in \mathbb{R}^{N \times d_z}$. We then apply standard self-attention to $H$:

$$\widetilde{H} = \text{softmax}\left( \frac{(HW_q^g)(HW_k^g)^\top}{\sqrt{d_z}} \right) (HW_v^g), \quad (10)$$

where $W_q^g, W_k^g, W_v^g \in \mathbb{R}^{d_z \times d_z}$ are learnable projection matrices. The attended sequence $\widetilde{H} \in \mathbb{R}^{N \times d_z}$ thus encodes explicit global interactions among all areas through a compact set of area-level embeddings. , although each axial block only attends along one axis of the OD grid, stacking multiple axial attention blocks gradually propagates information across origins and destinations. We

further apply more global-attention layers to $\widetilde{H}$, providing an additional low-resolution channel for information exchange across the entire region. This global-attention pathway strengthens the effect of the axial blocks by allowing all areas to attend to each other directly at the aggregated level.

We treat the output of the global attention blocks as an area-level representation, which contains origin-level and destination-level context tensors. GeoFlow then unsqueezes the area-level output along the origin and destination axes to form origin-aligned and destination-aligned views, and expands it back to the $N \times N$ OD grid via an outer-product-style broadcast so that each OD pair receives the corresponding origin- and destination-conditioned context. Finally, GeoFlow pass the resulting OD-level features through an MLP to obtain the final output. Fig. 1 upper right illustrates the axial and global attention mechanisms. Through this global-attention pathway, all areas interact explicitly via a compact set of pooled representations, enabling GeoFlow to capture high-order, long-range, and origin– and destination-crossing dependencies while keeping the computational and memory cost manageable.

### 3.3 OD FLOW PREDICTION AND GENERATION

GeoFlow provides an expressive, geometry-aware representation of intra-region area relationships that preserves multi-area spatial and relational cues necessary for accurate OD modeling. We apply GeoFlow to both OD flow prediction and generation tasks.

**Prediction.** For the prediction task, we append an MLP to the decoder output to produce the predicted OD flow matrix. Let $Z^{\mathrm{dec}} \in \mathbb{R}^{N \times N \times d_z}$ denote the decoder output; we flatten and project it via MLP $\phi_p : \mathbb{R}^{d_z} \to \mathbb{R}$, as

$$\hat{F} = \phi_p(Z^{\mathrm{dec}}) \tag{11}$$

where $\hat{F} \in \mathbb{R}^{N \times N}$ is the predicted OD flow matrix. The prediction loss is the average element-wise squared error

$$\mathcal{L}_{\mathrm{pred}} = \frac{1}{N^2} \left\| \hat{F} - F \right\|_2^2, \tag{12}$$

where $F$ is the ground-truth OD flow matrix and $|\cdot|_2$ is the Frobenius norm.

**Generation.** For the generation task, we adopt continuous-time flow matching (Lipman et al., 2022) conditioned on GeoFlow's fused embeddings. For each OD cell, we embed the instantaneous scalar $x_t[i, j]$ via an extra MLP encoder $\psi$ and fuse it with geometry and intrinsic embeddings:

$$Z_t[i, j, :] = \phi_f\big([E_g[i, j, :] \,\|\, \widetilde{E_x}[i, :] \,\|\, \widetilde{E_x}[j, :] \,\|\, \psi(x_t[i, j])]\big). \tag{13}$$

The decoder is followed by an MLP to predict the velocity field $v_\theta(x_t, t; Z_t) \in \mathbb{R}^{N \times N}$, similar to Eq. 11. The model is trained by mean squared error against the analytic target velocity induced by a perturbation schedule. The loss function can be written as

$$\mathcal{L}_{\mathrm{gen}} = \mathbb{E}_{t, x_0, \xi} \big\| v_\theta(x_t, t; Z_t) - v_t^{\mathrm{target}}(x_t \mid x_0) \big\|_2^2. \tag{14}$$

For a common linear–Gaussian schedule $x_t = \alpha(t)x_0 + \sigma(t)\xi$, the target velocity equals $\dot{\alpha}(t)x_0 + \dot{\sigma}(t)\xi$. At inference, we integrate the learned ordinary differential equation $\mathrm{d}x_t/\mathrm{d}t = v_\theta(x_t, t; Z_t)$ from a simple base $x_{t=0}$ to obtain samples. This formulation keeps training stable, conditions sampling on geometry and area identity, and yields faster inference than multi-step denoising methods. More details refer to Appendix A.1.

## 4 EXPERIMENTS

### 4.1 SETUP

**Objectives.** We aim to address three core questions through our experiments: (1) *How does GeoFlow perform compared to existing methods?* To this end, we design a suite of comprehensive metrics to systematically evaluate GeoFlow against competitive baselines in Sec. 4.2. (2) *Are the individual components of GeoFlow effective?* We conduct targeted ablation studies to verify the contribution of each component in Sec. 4.3. (3) *Is the design motivation of GeoFlow well-founded?* We perform a series of analysis experiments to demonstrate that our observations and motivations are not merely intuitive but also supported by empirical evidence in Sec. 4.4.

Table 1: Quantitative comparison on OD flow prediction (upper) and generation (lower) tasks. Results are averaged over three independent splits. ↑ indicates higher is better, and ↓ indicates lower is better. GeoFlow achieves the best overall performance.

| | CPC ↑ | RMSE↓ | MAE ↓ | JSD$_{in}$ ↓ | JSD$_{out}$ ↓ | JSD$_{all}$ ↓ | Div. ↑ |
|---|---|---|---|---|---|---|---|
| Gravity Model (Zipf, 1946) | 0.287 | 279.2 | 201.7 | 0.680 | 0.542 | 0.575 | - |
| RF (Pourebrahim et al., 2019) | 0.403 | 162.3 | 135.0 | 0.448 | 0.443 | 0.300 | - |
| GBRT (Robinson & Dilkina, 2018) | 0.401 | 147.8 | 121.1 | 0.434 | 0.423 | 0.316 | - |
| SVR (Rodríguez-Rueda et al., 2021) | 0.369 | 151.1 | 131.1 | 0.441 | 0.468 | 0.353 | - |
| DGM (Simini et al., 2021) | 0.383 | 145.9 | 106.9 | 0.463 | 0.464 | 0.312 | - |
| GMEL (Liu et al., 2020) | 0.380 | 156.4 | 114.2 | 0.467 | 0.299 | 0.297 | - |
| TransFlower (Luo et al., 2024) | 0.486 | 106.4 | 73.10 | 0.346 | 0.297 | 0.281 | - |
| GeoFlow$_{prediction}$ (Ours) | **0.604** | **85.64** | **54.83** | **0.238** | **0.131** | **0.171** | - |
| NetGAN (Bojchevski et al., 2018) | 0.405 | 142.0 | 99.35 | 0.439 | 0.312 | 0.252 | 0.070 |
| DiffODGen (Rong et al., 2023) | 0.464 | 159.9 | 80.50 | 0.331 | 0.228 | 0.204 | 0.058 |
| WEDAN (Rong et al., 2025) | 0.527 | 121.1 | 64.35 | **0.303** | 0.219 | 0.196 | 0.061 |
| GeoFlow$_{generation}$ (Ours) | **0.566** | **101.2** | **60.50** | 0.328 | **0.178** | **0.174** | **0.077** |

**Dataset.** We use the CommutingODGen dataset, a large-scale open-source benchmark for OD flow–related tasks (Rong et al., 2025). It contains commuting flow data covering 3,333 areas across diverse urban environments around the United States and provides a benchmark for OD flow prediction and generation methods. Although the dataset focuses on commuting flows, GeoFlow is designed in a general manner and can naturally extend to OD flows of other purposes.

**Metrics.** Since GeoFlow is evaluated on both OD flow prediction and generation, we adopt a set of metrics from multiple perspectives. (1) **Numerical accuracy.** For the prediction task, we employ Common Part of Commuting (CPC), Root Mean Square Error (RMSE), and Mean Absolute Error (MAE) to quantify the alignment between predicted flows and ground truth values. (2) **Distributional similarity.** Following prior work (Rong et al., 2023; 2025), we use Jensen–Shannon Divergence (JSD) to measure the discrepancy between the generated and true distributions of inflow, outflow, and OD flows. (3) **Diversity.** For the generation task, in addition to reconstruction accuracy and distributional similarity, we adopt the sample-level dissimilarity to assess diversity (Div.) across generated samples, which helps prevent potential mode collapse. Please refer to Appendix B.1 for more detailed information on metrics.

### 4.2 COMPARATIVE EVALUATION

**Baselines.** We compare GeoFlow against a broad and representative set of baselines: the classical gravity model; machine-learning regressors including RF, GBRT, and SVR; deep-learning approaches DGM, GMEL, and TransFlower; and recent graph-based methods NetGAN, DiffODGen, and WEDAN. More details and discussion about the baselines are provided in Appendix G.

**Protocol.** To ensure robustness and reproducibility, all methods are evaluated on three independent train/validation/test splits produced with different random seeds, and reported results are the arithmetic mean across these splits. For fair comparison, we standardize the search for hyperparameters (e.g., learning rate, batch size, hidden dimension, etc.) across methods.

**Results and analysis.** Quantitative results are presented in Tab. 1. Overall, GeoFlow achieves the best performance on both prediction and generation tasks. Consistent with prior studies, deep models (e.g., GMEL, TransFlower) outperform classical and shallow machine-learning baselines in predictive accuracy, while graph-based approaches (e.g., DiffODGen, WEDAN) deliver further gains by learning structured representations over the area network. GeoFlow attains additional improvements by explicitly incorate crucial geospatial features and employing specialized encoding and OD-aware decoding strategies, achieving a huge relative increase in CPC (0.604 vs. 0.486; 0.566 vs. 0.527) and improvement on RMSE (85.64 vs. 106.4) and MAE (54.83 vs. 73.10). In the generation setting, GeoFlow combines with a flow-matching module and consistently attains lower JSD for inflow/outflow/OD distributions (0.328/0.178/0.174 vs. 0.303/0.219/0.196), reduced reconstruction error, and higher sample-level diversity, indicating a better fit to the target distribution

Table 2: **Ablation study on input features.** SL, Rel., FF represent straight-line distance, relative coordinates, and free flow (geodesic) distance, respectively.

| ID | SL | Rel. | $k$-hop | FF | CPC↑ | RMSE↓ | JSD↓ |
|----|----|------|---------|----|------|-------|------|
| 1 | ✓ | | | | 0.462 | 196.8 | 0.252 |
| 2 | ✓ | | ✓ | ✓ | 0.515 | 144.5 | 0.230 |
| 3 | ✓ | ✓ | | ✓ | 0.514 | 184.0 | 0.224 |
| 4 | ✓ | ✓ | ✓ | | 0.510 | 170.4 | 0.222 |
| 5 | ✓ | ✓ | ✓ | ✓ | **0.527** | **130.9** | **0.194** |

Table 3: **Ablation study on encoder and decoder architectures.** Enc. and Dec. represent the proposed enhanced encoder and decoder, respectively.

| ID | Enc. | Dec. | CPC↑ | RMSE↓ | JSD↓ |
|----|------|------|------|-------|------|
| 6 | | | 0.510 | 226.5 | 0.209 |
| 7 | ✓ | | 0.514 | 171.3 | 0.217 |
| 8 | | ✓ | 0.521 | 160.9 | 0.220 |
| 9 | ✓ | ✓ | **0.527** | **130.9** | **0.194** |

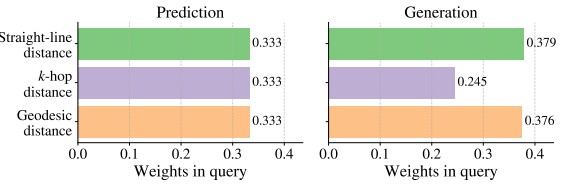

Figure 2: **Weights of geospatial feature in intrinsic attributes aggregation.** The augmented geospatial features are highly attended. Complementarity of different distances is further discussed in Sec. 4.4.

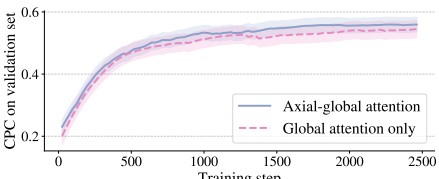

Figure 3: **CPC trends during training.** The axial–global attention demonstrates more stable convergence and consistently outperforms the global-only one.

while capturing more diverse mobility patterns and mitigating mode collapse. Further fine-grained analyses of performance across region sizes and generalizability are provided in Appendix C.1.

### 4.3 ABLATION STUDIES

**Ablation on geospatial feature augmentation.** We conduct an ablation study on augmented geospatial features by isolating the effects of relative position (ID 2), $k$-hop distance (ID 3), and geodesic distance (ID 4), benchmarked against a baseline using only straight-line distance (ID 1). As shown in Tab. 2, the full model achieves the best performance (ID 5), with augmented features consistently improving over the baseline. Relative position encodes the placement of one area relative to another under a fixed regional orientation, providing essential cues for locational context and multi-area geometric dependencies. The $k$-hop distance and free-flow travel time capture network-based accessibility and structural constraints that strongly influence mobility and commuting behavior. Beyond individual features, we examine their contribution to the intrinsic attribute aggregation in Eq. 4. Fig. 2 shows that straight-line distance, $k$-hop distance, and geodesic distance receive substantial contribution weights in prediction and generation tasks, highlighting the critical role of geospatial attributes in modeling inter-area interactions. In prediction, the weights across all features are nearly uniform; however, this does not indicate redundancy between augmented and original features. To clarify this, we further perform a fine-grained similarity analysis among geospatial features, detailed in Sec. 4.4.

**Ablation on encoder and decoder architecture.** We ablate the proposed encoder for geometric–intrinsic attributes fusing and the axial–global attention decoder. Without the proposed encoder, we replace it with a conventional graph convolution (GC) module (ID 6 and 8); without the decoder, we substitute a vanilla global attention mechanism of equal depth (ID 6 and 7). As reported in Tab. 3, the full encoder–decoder configuration achieves the best performance (ID 9). On the encoder side, conventional GC is insufficient to exploit geospatial attributes, while the proposed encoder, by explicitly incorporating them (see Fig. 2), produces more discriminative and coherent embeddings. On the decoder side, axial–global attention captures both destination preferences and reciprocal capacity effects, modeling diverse OD flow patterns with higher fidelity. Fig. 3 further shows that axial–global attention improves training stability and predictive accuracy over global-only attention, demonstrating that it reduces computational cost while sustaining or enhancing performance under comparable budgets. In addition, we conduct ablation experiments comparing diffusion and flow-matching models, with details provided in Appendix C.3.

Table 4: Similarity analysis among geographic information. F. norm represents the Frobenius norm, and MAD represents the maximal absolute difference. Each matrix is normalized to row-sum to one.

|  | SL vs. $k$-hop | SL vs. FF | $k$-hop vs. FF |
|---|---|---|---|
| F. norm | 8.07 | 0.741 | 7.98 |
| MAE | 0.149 | 0.021 | 0.146 |
| MAD | 0.424 | 0.128 | 0.407 |

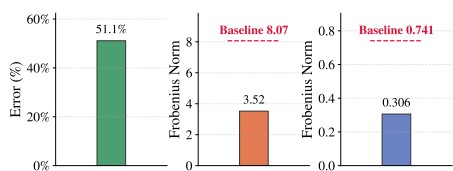

Figure 4: Prediction errors when inferring higher-order geospatial attributes from adjacency and straight-line distance matrices.

### 4.4 ANALYSIS EXPERIMENTS

**Complementarity of straight-line and network-based distances.** We examine whether $k$-hop and free-flow (FF; i.e., geodesic) distances provide information beyond straight-line (SL) distance by comparing their similarities. Each distance matrix is origin-normalized so rows sum to one, removing scale-induced bias. The results are summarized in Tab. 4. Consistent with intuition, areas close in SL distance are generally close in $k$-hop and FF distances, yielding similar overall distributions. Quantitatively, however, agreement varies: SL–FF shows relatively small discrepancies (0.741 Frobenius norm, 0.021 MAE), whereas comparisons with $k$-hop yield much larger values ($\approx 8.0$ Frobenius norm, 0.15 MAE). Even SL–FF has a non-negligible maximal absolute difference (MAD) of 0.128, indicating accumulated errors for long-range pairs. These results suggest that SL distance alone may miss long-range distortions, underscoring the need to incorporate network-based descriptors such as $k$-hop or FF distances when modeling geospatial dependencies.

**Difficulty of inferring higher-order geospatial attributes.** Although adjacency and distance matrices theoretically encode complete spatial information, empirically recovering higher-order attributes from them is challenging. To quantify this, we train an MLP to approximate relative angle, $k$-hop distance, and geodesic (free-flow) distance from the raw matrices. As shown in Fig. 4, relative-angle prediction yields an average relative error of 51.1%, and the reconstructed $k$-hop and geodesic matrices exhibit large Frobenius-norm discrepancies, though somewhat smaller than those reported in Tab. 4. These results indicate that models struggle to recover shortest-path and angular structure solely from adjacency and straight-line distance information. This difficulty is further compounded in the OD-flow setting, where task complexity degrades the recovery of critical geospatial cues. By contrast, directly providing such features, which are readily obtainable through simple preprocessing, offers a straightforward and effective remedy.

## 5 CONCLUSION

In this work, we introduce GeoFlow, a novel framework for modeling origin–destination (OD) flows. GeoFlow augments critical yet often overlooked geospatial relationships as representations and employs a geometric–intrinsic fusion encoder coupled with an axial–global attention decoder, achieving state-of-the-art performance on both prediction and generation tasks. Extensive experiments validate the effectiveness of the proposed approach. Ablation and analytical studies further highlight the significance of geospatial attributes and the encoder–decoder design, offering new insights for OD flow modeling and related applications.

### REPRODUCIBILITY STATEMENT

We take measures to ensure reproducibility. Methods, implementation, and experiments are described in detail in the main text and appendix. We use publicly accessible datasets and benchmarks and release our source code. To reduce variability from dataset splitting and training, we run experiments with fixed random seeds. Accordingly, the reported results should be reproduced.

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

# APPENDIX

## A   METHOD DETAILS

### A.1   FLOW MATCHING MODELS FOR ORIGIN-DESTINATION FLOW GENERATION

To generate high-fidelity origin-destination (OD) flow matrices, we adopt Flow Matching (FM), an efficient and expressive framework for generative modeling (Lipman et al., 2022; 2024). In contrast to diffusion models, which typically require simulating stochastic or ordinary differential equations (ODEs) during training, FM directly learns the vector field that transports a simple prior distribution to the complex data distribution. This simulation-free paradigm substantially improves training efficiency.

#### A.1.1   FORMULATION

The core principle of FM is to define a continuous-time flow from a simple prior distribution $p_0(\mathbf{x}_0)$, typically a standard normal distribution $\mathcal{N}(\mathbf{0}, \mathbf{I})$, to the target data distribution $p_1(\mathbf{x}_1)$, which in our case corresponds to OD matrices. This flow is governed by an ODE:

$$\frac{d\mathbf{x}_t}{dt} = v_t(\mathbf{x}_t), \tag{15}$$

where $t \in [0, 1]$ is a pseudo-time variable, $\mathbf{x}_t$ denotes the state at time $t$, and $v_t$ is a time-dependent vector field. The goal is to train a neural network $v_\theta(\mathbf{x}_t, t)$ to approximate this vector field.

To derive a tractable training objective, we first define a conditional probability path $p_t(\mathbf{x}|\mathbf{x}_1)$ that connects a prior sample $\mathbf{x}_0 \sim p_0(\mathbf{x}_0)$ to a data sample $\mathbf{x}_1 \sim p_1(\mathbf{x}_1)$. Following Lipman et al. (2022), we adopt a simple linear interpolation path:

$$\mathbf{x}_t = (1 - t)\mathbf{x}_0 + t\mathbf{x}_1, \tag{16}$$

with the corresponding conditional vector field:

$$u_t(\mathbf{x}_t|\mathbf{x}_1) = \mathbf{x}_1 - \mathbf{x}_0. \tag{17}$$

For this linear path, the target vector field is constant and independent of both $\mathbf{x}_t$ and $t$.

The FM objective is to minimize the expected mean squared error between the predicted vector field $v_\theta$ and the target vector field $u_t$:

$$\mathcal{L}_{\text{FM}} = \mathbb{E}_{t\sim\mathcal{U}(0,1),\mathbf{x}_0\sim p_0(\mathbf{x}_0),\mathbf{x}_1\sim p_1(\mathbf{x}_1)} \left[ \|v_\theta((1 - t)\mathbf{x}_0 + t\mathbf{x}_1, t) - (\mathbf{x}_1 - \mathbf{x}_0)\|^2 \right]. \tag{18}$$

#### A.1.2   TRAINING

In practice, optimization proceeds by sampling mini-batches. For each training step, a data point $\mathbf{x}_1$ (an OD matrix), a noise vector $\mathbf{x}_0 \sim \mathcal{N}(\mathbf{0}, \mathbf{I})$, and a time step $t \sim \mathcal{U}(0, 1)$ are sampled. The interpolated state $\mathbf{x}_t$ and target vector $\mathbf{v}_{\text{target}} = \mathbf{x}_1 - \mathbf{x}_0$ are then computed. The model $v_\theta$ takes $\mathbf{x}_t$, $t$, and associated geospatial context as input to predict a velocity $\mathbf{v}_{\text{pred}}$, and the mean squared error between $\mathbf{v}_{\text{pred}}$ and $\mathbf{v}_{\text{target}}$ is minimized via backpropagation.

#### A.1.3   SAMPLING

Once trained, $v_\theta$ enables the generation of new OD matrices by solving the initial value problem defined by the learned ODE. Starting from a prior sample $\mathbf{x}_0 \sim \mathcal{N}(\mathbf{0}, \mathbf{I})$, we integrate the ODE from $t = 0$ to $t = 1$ using a numerical solver. In our implementation, the `sample` method employs the forward Euler scheme:

$$\mathbf{x}_{t+\Delta t} = \mathbf{x}_t + v_\theta(\mathbf{x}_t, t)\Delta t, \tag{19}$$

where $\Delta t$ is the integration step size. After $N_{\text{steps}} = 1/\Delta t$ iterations, the resulting state $\mathbf{x}_1$ constitutes a novel sample drawn from the learned distribution. This procedure is executed independently for each graph in a batch, enabling efficient generation of multiple OD matrices.

## A.2 PREPROCESSING

In this work, GeoFlow is trained and evaluated on the CommutingODGen dataset (Rong et al., 2025), a collection of commuting OD flows around urban areas, where each area is represented as a graph in matrix form. The nodes correspond to geographic zones (e.g., census tracts), and the edges represent road network adjacency and Euclidean distance. For each area, the dataset provides node attributes, spatial coordinates, a road network adjacency matrix, a Euclidean distance matrix, and the ground-truth commuting OD flow matrix. A standardized preprocessing pipeline is applied to harmonize these inputs and derive multi-view relational structures.

The preprocessing for each urban area consists of the following steps:

1. **Area Feature Scaling:** For each area, we construct a raw feature vector by concatenating demographic attributes with counts of points of interest (POIs). To ensure comparability across features, we apply column-wise max normalization, dividing each feature value by the maximum observed within the same area. This rescales all features to the range $[0, 1]$.

2. **Coordinate Normalization:** The raw geographic coordinates (zone centroids) are standardized to obtain a canonical representation invariant to translation and rotation. Specifically, we first subtract the mean to center the coordinates. We then apply Principal Component Analysis (PCA) (Abdi & Williams, 2010) to align the coordinate axes with directions of maximal variance. Finally, the transformed coordinates are rescaled to the range $[-1, 1]$ by dividing by the maximum absolute value along each axis.

3. **OD Flow Normalization:** The raw OD matrix, containing integer-valued flows, is normalized to form a probability distribution by dividing by the total commuting volume of the area. This total volume is stored and later used during inference to rescale generated probability matrices back to absolute flows.

4. **Multi-view Affinity Construction:** To capture diverse spatial relationships between zones, we construct three affinity matrices from different distance measures. Given a distance matrix $\mathbf{D}$, we compute an affinity matrix $\mathbf{A}$ using an exponential kernel:

$$\mathbf{A}_{ij} = \exp(-\mathbf{D}_{ij}/\tau), \tag{20}$$

where $\tau$ is set to the median of all positive, finite distances in $\mathbf{D}$. This produces a dense, weighted graph where affinity decreases with distance. The three distance notions are:

- **Euclidean Affinity:** Based on straight-line distance between zone centroids, capturing pure spatial proximity.
- **Topological Affinity:** Based on unweighted shortest-path distance (number of hops) on the road network, reflecting connectivity in the transport infrastructure.
- **Network-Geodesic Affinity:** Based on shortest-path distance on the road network weighted by edge lengths, representing the most efficient travel distance along the network.

For each affinity matrix, we also compute its symmetrically normalized form,

$$\mathbf{A}_{\mathrm{norm}} = \mathbf{D}_{\mathrm{deg}}^{-1/2} \mathbf{A} \mathbf{D}_{\mathrm{deg}}^{-1/2}, \tag{21}$$

where $\mathbf{D}_{\mathrm{deg}}$ is the weighted generalized degree matrix with diagonal entries $(\mathbf{D}_{\mathrm{deg}})_{ii} = \sum_j \mathbf{A}_{ij}$, i.e., the total connection strength of node $i$. Symmetric normalization mitigates scale differences introduced by heterogeneous node degrees and yields affinity matrices that are numerically stable and comparable across areas. We include all normalized affinity matrices in the geometric relation descriptor $G$, which supplies multi-relational contexts to downstream modules.

## B EXPERIMENT DETAILS

### B.1 METRIC DETAILS

We provide the formal definitions and additional details of the evaluation metrics in this section.

**Numerical Accuracy.** To evaluate OD flow prediction and reconstruction, we employ three standard metrics:

- **Common Part of Commuting (CPC).** CPC measures the similarity between two OD matrices by computing the fraction of overlapping flow volume:

$$\text{CPC}(\mathbf{X}, \mathbf{Y}) = \frac{2 \sum_{i,j} \min(X_{ij}, Y_{ij})}{\sum_{i,j} X_{ij} + \sum_{i,j} Y_{ij}}, \tag{22}$$

where $\mathbf{X}$ and $\mathbf{Y}$ denote the predicted and ground-truth OD matrices, respectively. Higher values indicate better alignment.

- **Root Mean Square Error (RMSE).** RMSE quantifies the average magnitude of error in flow counts:

$$\text{RMSE}(\mathbf{X}, \mathbf{Y}) = \sqrt{\frac{1}{N^2} \sum_{i,j} (X_{ij} - Y_{ij})^2}. \tag{23}$$

- **Mean Absolute Error (MAE).** MAE provides a scale-independent measure of average deviation:

$$\text{MAE}(\mathbf{X}, \mathbf{Y}) = \frac{1}{N^2} \sum_{i,j} |X_{ij} - Y_{ij}|. \tag{24}$$

**Distributional Similarity.** To assess whether generated flows capture realistic distributional patterns, we compute the Jensen–Shannon divergence (JSD) between the distributions of inflows, outflows, and OD entries:

$$\text{JSD}(P\|Q) = \frac{1}{2}\text{KL}(P\|M) + \frac{1}{2}\text{KL}(Q\|M), \tag{25}$$

where $M = \frac{1}{2}(P + Q)$ and $\text{KL}(\cdot\|\cdot)$ denotes the Kullback–Leibler divergence. JSD is symmetric and bounded between 0 and 1, with lower values indicating closer distributions.

**Diversity.** For the generation task, we evaluate sample diversity using the metric defined as

$$\text{Diversity}(\{\tilde{\mathbf{F}}^k\}_{k=1}^K) = \frac{2}{K(K-1)} \sum_{k<\ell} d(\tilde{\mathbf{F}}^k, \tilde{\mathbf{F}}^\ell), \tag{26}$$

where $\{\tilde{\mathbf{F}}^k\}_{k=1}^K$ are independently generated OD matrices under the same conditioning, and $d(\cdot, \cdot)$ is a distance measure (e.g., in this work, RMSE). This metric computes the average pairwise dissimilarity among generated samples, with higher values indicating greater diversity. Intuitively, a model affected by mode collapse produces more similar or even identical samples with uniformly sampled random seeds, leading to high similarity (low dissimilarity and diversity), whereas a well-generalized model generates sufficiently distinct OD matrices, resulting in lower similarity and higher diversity.

## B.2 TRAINING DETAILS

GeoFlow is trained end-to-end using the AdamW optimizer with an initial learning rate of $1 \times 10^{-3}$ and a batch size of 64. The training objective is the Mean Squared Error (MSE) between the predicted and ground-truth normalized OD matrices. A ReduceLROnPlateau learning rate scheduler is employed, which decreases the learning rate by a factor of 0.5 if the validation metric does not improve for 10 consecutive epochs. To mitigate overfitting, early stopping is applied with a patience of 100 epochs. Model performance is evaluated on the validation set after each epoch, and the checkpoint achieving the best validation performance is retained. All experiments are implemented in PyTorch and conducted on a single NVIDIA GeForce RTX 4090 GPU.

## B.3 HYPERPARAMETERS

The model architecture and training configuration are controlled by a set of specified hyperparameters. The hidden dimension for geometric and intrinsic embeddings is set to 32, which we found to provide a favorable balance between computational efficiency and predictive accuracy. The encoder consists of two multilayer perceptron (MLP) layers for encoding geometric attributes and fusing geometric and intrinsic attributes, followed by two graph attention layers. The decoder is composed of two layers of axial self-attention and two layers of global self-attention. All attention modules employ four heads, and dropout is disabled throughout training.

Table 5: Partition of regions by size.

| Region Size | # Areas | # Regions |
|---|---|---|
| Small | $< 10$ | 1080 |
| Medium | $10 \leq, < 100$ | 1023 |
| Large | $\geq 100$ | 149 |
| All | - | 2252 |

Table 6: Results across training–evaluation scale pairs. IDs indicate the different configurations of training and evaluation set composition.

| ID | Training Region Size | | | Evaluate on | Metrics | | | | | |
|---|---|---|---|---|---|---|---|---|---|---|
| | Small | Medium | Large | Region Size | CPC ↑ | RMSE ↓ | MAE ↓ | $JSD_{in}$ ↓ | $JSD_{out}$ ↓ | $JSD_{all}$ ↓ |
| 1 | ✓ | ✓ | ✓ | All | 0.630 | 72.95 | 47.09 | **0.231** | 0.126 | 0.158 |
| 2 | | ✓ | ✓ | | 0.587 | 83.29 | 51.89 | 0.279 | 0.210 | 0.166 |
| 3 | ✓ | | ✓ | | 0.575 | 93.46 | 57.99 | 0.270 | 0.160 | 0.226 |
| 4 | ✓ | ✓ | | | **0.647** | **64.91** | **41.63** | 0.241 | **0.096** | **0.141** |
| 5 | ✓ | ✓ | ✓ | Small | 0.769 | 72.62 | 49.68 | 0.244 | 0.099 | 0.136 |
| 6 | | ✓ | ✓ | | 0.638 | 108.4 | 67.31 | 0.349 | 0.292 | 0.182 |
| 7 | ✓ | | ✓ | | 0.729 | 85.68 | 60.44 | 0.286 | 0.104 | 0.173 |
| 8 | ✓ | ✓ | | | **0.785** | **66.26** | **46.21** | 0.238 | **0.087** | **0.134** |
| 9 | ✓ | ✓ | ✓ | Medium | 0.559 | 62.39 | 40.12 | 0.202 | 0.119 | 0.154 |
| 10 | | ✓ | ✓ | | **0.597** | **55.30** | **35.36** | **0.201** | 0.095 | 0.130 |
| 11 | ✓ | | ✓ | | 0.484 | 96.23 | 58.55 | 0.240 | 0.196 | 0.261 |
| 12 | ✓ | ✓ | | | 0.579 | 58.00 | 36.15 | 0.205 | **0.086** | **0.124** |
| 13 | ✓ | ✓ | ✓ | Large | 0.139 | 102.62 | 46.06 | 0.504 | **0.234** | 0.305 |
| 14 | | ✓ | ✓ | | **0.150** | **93.32** | 53.70 | **0.310** | 0.401 | **0.301** |
| 15 | ✓ | | ✓ | | 0.081 | 130.90 | **36.33** | 0.360 | 0.315 | 0.372 |
| 16 | ✓ | ✓ | | | 0.115 | 147.9 | 76.15 | 0.334 | 0.362 | 0.343 |

To ensure robustness, we conducted sensitivity analyses over different hidden dimensions, numbers of encoder/decoder layers, and attention heads. The reported configuration consistently achieved the best trade-off between performance and efficiency across validation runs, and is therefore adopted in the main experiments.

## C  ADDITIONAL EXPERIMENTS

### C.1  GENERALIZABILITY

Beyond the main-text evaluations, we further probe the generalizability of GeoFlow along three complementary axes. First, in the prediction setting, we vary the spatial scale of training and evaluation regions to quantify how well the model transfers across omitted area scales. Second, in the generation setting, we construct longitudinal partitions of the study area and test whether a model trained on one part of the space can faithfully reproduce patterns in another. Finally, we perform a fine-grained performance analysis by conditioning on OD-flow magnitude and OD distance, which reveals how error is distributed across practically relevant regimes.

#### C.1.1  GENERALIZABILITY ACROSS AREA SCALES

To better understand how predictive performance and transferability vary with the spatial scale of regions, we adopt a scale-aware evaluation protocol that explicitly varies both the region sizes present during training and the region sizes on which models are evaluated.

Table 7: Generalizability across longitudinal partitions. "All – West" (resp. "All – East") denotes that the model is trained on all regions except those in the held-out western (resp. eastern) partition. "All – Random" uses a randomly sampled held-out set with the same cardinality.

| ID | Train on | Evaluate on | CPC↑ | RMSE↓ | JSD↓ |
|----|----------|-------------|------|-------|------|
| 1 | All - West | West | 0.572 | 111.1 | 0.176 |
| 2 | All - East | East | 0.561 | 116.7 | 0.177 |
| 3 | All - Random | Random | 0.566 | 101.2 | 0.174 |

**Setup.** For the experiments, regions are partitioned by the number of constituent areas into three size classes: small, medium, and large, as summarized in Tab. 5. We train distinct models under a set of controlled training compositions formed by varying the inclusion and exclusion of these size classes (for example, ID 4 is trained on the union of small- and medium-sized regions). Each trained model is evaluated separately on small, medium, large, and on the union of all regions (All). This design isolates how particular training mixtures influence performance both within and across scales and enables a principled comparison of generalizability under differing class imbalances and distributional shifts.

**Analysis.** When the evaluation scale is absent from the training pool (IDs 6, 11, and 16), performance on the held-out class deteriorates consistently. RMSE and MAE increase, and CPC declines, which confirms that excluding a scale impairs direct transfer. Crucially, however, the magnitude of this degradation is modest relative to a model trained on the full corpus (IDs 5, 9, and 13). In other words, although removing the target scale produces a measurable loss, the model nevertheless retains a substantial portion of its predictive ability across scales, indicating that it learns components of OD structure that generalize beyond the observed size class.

### C.1.2 GENERALIZABILITY ACROSS LONGITUDINAL PARTITIONS

To complement the above scale-based study, we evaluate generalizability under a different kind of spatial shift in the generation setting. We partition the study areas into eastern and western subsets based on longitude and additionally construct a random partition with the same cardinality as a baseline.

**Setup.** For the "All - West" configuration, the generative model is trained on all regions except those in the western subset and then evaluated exclusively on the held-out West partition. "All - East" and "All - Random" are defined analogously. We report CPC, RMSE, and JSD between the generated and ground-truth OD distributions on the corresponding held-out partitions in Tab. 7.

**Analysis.** Across all three configurations, performance remains stable. CPC stays at a similar level across splits, RMSE differences are moderate, and JSD exhibits minor fluctuations. The random partition does not confer a clear advantage over structured East/West splits, suggesting that the model does not overfit to idiosyncratic geographic patterns of any particular longitudinal band. Instead, it captures generative regularities that transfer across distinct spatial subdomains.

### C.1.3 PERFORMANCE ANALYSIS

**Scale-dependent predictive difficulty.** This partitioning makes explicit two salient characteristics of the dataset: (1) small and medium regions dominate numerically, whereas large regions are comparatively scarce; and (2) the intrinsic complexity of OD structure grows with region size. Consistent with these observations, as shown in Tab. 6, predictive difficulty exhibits a stable ordering: models attain the best results on small regions, intermediate results on medium regions, and the worst results on large regions. Large-region evaluations display substantially higher RMSE and lower CPC, indicating both greater pointwise error and poorer fidelity of the learned OD distributions. This pattern is interpretable as the consequence of increased intra-region heterogeneity and multi-modal travel structure in larger spatial extents.

**Performance profile across flow scales.** Tab. 8 conditions performance on OD-flow magnitude. The distribution is highly skewed. More than $83\%$ of OD pairs have flows below 5, whereas flows above 100 account for less than $1\%$ of the pairs. As flow increases, absolute RMSE naturally grows,

Table 8: Performance breakdown by OD flow magnitude. Ratio denotes the proportion of OD pairs falling into each flow bin, and Inv. Areas is the fraction of areas that participate in at least one interaction within the bin. NRMSE is the RMSE normalized within absolute flow magnitude.

| ID | Flow Scale | Proportion | Inv. Areas | CPC ↑ | RMSE ↓ | NRMSE ↓ |
|----|-----------|-----------|-----------|-------|--------|---------|
| 1 | $0 \leq, < 5$ | 83.18% | 85.84% | 0.105 | 146.1 | 219.1 |
| 2 | $5 \leq, < 20$ | 11.62% | 96.46% | 0.269 | 185.7 | 19.56 |
| 3 | $20 \leq, < 100$ | 4.49% | 100.0% | 0.493 | 256.1 | 6.404 |
| 4 | $100 \leq, < 500$ | 0.69% | 99.71% | 0.620 | 373.9 | 2.063 |
| 5 | $500 \leq, < 1000$ | 0.02% | 48.08% | 0.640 | 665.9 | 1.036 |
| 6 | $\geq 1000$ | 0.01% | 8.55% | 0.513 | 1330 | 1.010 |

Table 9: Performance breakdown by normalized OD distance. Ratio and Inv. Areas are defined as in Tab. 8.

| ID | Distance Scale | Proportion | Inv. Areas | CPC ↑ | RMSE ↓ | NRMSE ↓ |
|----|---------------|-----------|-----------|-------|--------|---------|
| 1 | $0 \leq, < 0.1$ | 4.67% | 100.0% | 0.556 | 447.9 | 22.56 |
| 2 | $0.1 \leq, < 0.2$ | 8.66% | 49.85% | 0.418 | 389.2 | 46.27 |
| 3 | $0.2 \leq, < 0.5$ | 31.13% | 73.16% | 0.471 | 244.5 | 47.34 |
| 4 | $0.5 \leq, < 1$ | 40.33% | 95.28% | 0.483 | 163.4 | 55.51 |
| 5 | $1 \leq, < 2$ | 15.16% | 100.0% | 0.408 | 122.6 | 46.37 |
| 6 | $\geq 2$ | 0.05% | 39.82% | 0.251 | 64.62 | 8.626 |

but normalized error (NRMSE) drops and CPC improves from $0.105$ in the smallest bin to around $0.62$–$0.64$ for flows in $[100, 1000)$. This indicates that, in relative terms, the model is substantially more reliable on high-volume OD pairs that dominate aggregate mobility patterns, while low-flow interactions, which are both noisier and less constrained by data, remain the most challenging. Moreover, when baseline flows are very small, CPC becomes highly sensitive to minor absolute fluctuations: a handful of misallocated trips can induce large apparent drops in correlation, making the low-flow bins look disproportionately poorly modeled. This is consistent with the interpretation that OD pairs with very low counts contain a substantial random component, and such randomness, combined with a small base volume, translates into amplified variability of distributional metrics. The slight CPC decrease in the $\geq 1000$ bin is plausibly attributable to extreme data sparsity (only $0.01\%$ of pairs, involving $8.55\%$ of areas).

**Performance profile across distance scales.** We further examine performance as a function of normalized OD distance in Tab. 9. Most OD pairs correspond to short- to medium-range trips: distances in $[0.2, 1)$ already account for over 70% of the mass. CPC is highest for the shortest-range bin $[0, 0.1)$ and remains relatively stable for distances up to 1, indicating that the model captures local and intra-region mobility patterns well. For very long-range interactions ($\geq 2$), CPC drops markedly to $\approx 0.25$ despite a lower NRMSE, reflecting that these flows are extremely sparse (only $0.05\%$ of OD pairs, involving $39.82\%$ of areas) and concentrated along a small number of long-distance corridors. Overall, the distance-conditioned analysis shows that the model is most accurate in the distance regimes that dominate real-world demand, while rare long-distance trips and very short but noisy interactions remain more difficult to predict perfectly.

## C.2 SCALABILITY

To assess the scalability of our architecture, we systematically increase model capacity along two axes: width (hidden dimension) and depth (number of encoder/decoder attention layers and attention heads) with all other optimization hyperparameters and training schedules kept fixed.

Tab. 10 reports the results. Increasing the hidden dimension from 16 to 64 (IDs 1–3) yields consistent gains. CPC improves from $0.576$ to $0.593$, RMSE drops from $107.5$ to $92.43$, and JSD decreases from $0.212$ to $0.194$. This indicates that the model can effectively leverage additional representational capacity, and that the architecture behaves in a stable, roughly monotone way under width scaling. Adding moderate depth and more heads on top of a 32-dimensional backbone (ID 4) fur-

Table 10: Effect of model capacity on performance. We vary the hidden dimension and the number of encoder/decoder attention layers and attention heads, while keeping all other hyperparameters fixed. The number of attention layers is applied to both axial and global attention blocks.

| ID | Hidden Dimension | # Encoder Attn Layers | # Decoder Attn Layers | # Attn Heads | CPC↑ | RMSE↓ | JSD↓ |
|----|------------------|------------------------|------------------------|---------------|-------|--------|-------|
| 1  | 16 | 1 | 1 | 2 | 0.576 | 107.5 | 0.212 |
| 2  | 32 | 1 | 1 | 2 | 0.582 | 98.59 | 0.200 |
| 3  | 64 | 1 | 1 | 2 | 0.593 | 92.43 | 0.194 |
| 4  | 32 | 2 | 2 | 4 | 0.601 | 86.98 | 0.189 |

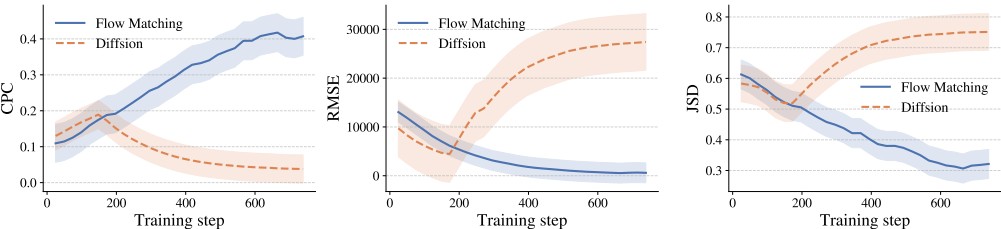

Figure 5: Training dynamics comparison between flow-matching and diffusion models. Flow-matching demonstrates steady improvements across metrics, while diffusion exhibits instability and performance degradation.

ther improves all metrics, reaching higher CPC and lower RMSE, and lower JSD. The incremental improvements suggest diminishing returns at higher capacities, consistent with a regime where performance is increasingly data-limited rather than model-limited. Overall, these results show that our model scales gracefully. Larger configurations provide better accuracy, but even the smallest variant (ID 1) attains a substantial fraction of the performance of the largest one. This makes it possible to trade off compute and memory for accuracy depending on deployment constraints, while retaining the same architectural design and training procedure.

### C.3  COMPARISON OF DIFFUSION MODELS AND FLOW MATCHING MODELS

To further assess the effectiveness of our framework, we conduct a comparative study between flow-matching and diffusion models. In this setting, only the flow-matching components are replaced with their diffusion counterparts, while all other configurations remain unchanged. Since diffusion models follow a different generative paradigm and training procedure, we perform extensive hyperparameter tuning to ensure comparable model size and computational cost. As shown in Fig. 5, although the diffusion model initially exhibits normal metric progression, it undergoes severe oscillations and sustained degradation. This instability arises from the inherently slower and noisier training dynamics of diffusion, which hinder efficient convergence. In contrast, the flow-matching model exhibits consistent and monotonic improvements throughout training, indicating its stability and efficiency in capturing the underlying OD flow structure.

## D  COMPUTATIONAL COMPLEXITY AND RESOURCE USAGE

We profile the computational footprint of our proposed GeoFlow architecture against the Trans-Flower baseline. Table 11 reports average training and inference wall-clock time, average and peak FLOPs, and peak GPU memory usage, computed using the same configuration as that used to obtain its main performance results reported in the main experiments. All measurements are taken from steady-state runs after warm-up, so the numbers reflect typical rather than best-case performance.

GeoFlow achieves noticeably shorter wall-clock times than TransFlower: training is roughly $3.6\times$ faster (0.0160 s vs. 0.0580 s) and inference is about $4\times$ faster (0.1104 s vs. 0.4468 s). At the same time, GeoFlow requires around $4\times$ higher average FLOPs ($6.12 \times 10^8$ vs. $1.49 \times 10^8$) and a corresponding increase in peak FLOPs ($3.58 \times 10^{11}$ vs. $8.82 \times 10^{10}$), while peak memory consump-

Table 11: Computational cost of GeoFlow architecture compared to the TransFlower baseline. We report average training and inference wall-clock time, average and peak floating-point operations (FLOPs), and peak GPU memory usage.

| | Training Time (s) | Inference Time (s) | Avg. FLOPs | Peak FLOPs | Peak Mem. (MiB) |
|---|---|---|---|---|---|
| TransFlower | 0.0580 | 0.4468 | $1.49 \times 10^8$ | $8.82 \times 10^{10}$ | $2.31 \times 10^4$ |
| GeoFlow | 0.0160 | 0.1104 | $6.12 \times 10^8$ | $3.58 \times 10^{11}$ | $2.23 \times 10^4$ |

tion remains comparable between the two models. This contrast highlights that GeoFlow adopts a more compute-intensive yet highly parallelizable design that leads to better hardware utilization and throughput, whereas TransFlower offers lower FLOP counts at the cost of higher latency. Taken together with the accuracy improvements of GeoFlow, these results indicate that GeoFlow provides a favorable accuracy–efficiency trade-off, while remaining tractable at city scale on a single general-purpose GPU.

## E    RECOMMENDATIONS FOR TRAINING-SET COMPOSITION

The generalizability experiments in Appendix C.1.1 suggest several practical guidelines for choosing which size classes to include when constructing the training set. The selection of region sizes has systematic and interpretable consequences for both within-scale accuracy and cross-scale transfer.

Models trained on the combination of small and medium areas tend to produce the strongest results for small-scale and aggregate evaluations (IDs 4, 8, 12 and 16 in Tab. 6) and, in aggregate metrics, can even outperform the model trained on the full corpus (ID 4 vs. ID 1). This apparent advantage is primarily attributable to the numerical predominance of small and medium areas: their abundance supplies stronger and more stable gradient signals during optimization, which reduces average error on the dominant scales and therefore improves aggregate statistics. However, the same training regimen is weaker on large-area evaluation than the model trained on all scales (ID 16 vs. ID 13). Including the relatively scarce but distributionally distinct large-area examples in the training set increases the model's ability to capture large-area specific structure and yields better performance on large areas and, importantly, stronger overall generalizability across all scales.

By contrast, training on medium and large areas is most effective when the evaluation target is medium or large (IDs 10 and 14), which suggests that medium areas provide transitional patterns that facilitate transfer toward large regimes. Naively pairing small and large areas without an intermediate representation often degrades performance (IDs 7, 11 and 15), likely because the omission of medium examples amplifies the domain shift between the two extremes.

In summary, the results point to a practical trade-off. Emphasizing small and medium areas during training tends to improve aggregate metrics and accuracy on the majority of areas in our corpus, whereas incorporating the relatively scarce large-area examples improves fidelity on large areas and enhances cross-scale robustness. This pattern appears consistently in our dataset but should be interpreted with caution rather than as a universal prescription. The balance between aggregate performance and coverage of data-scarce large areas is inherently dataset dependent and therefore calls for pragmatic, case-by-case choices when assembling training data in practical applications.

## F    QUALITATIVE EVALUATION

### F.1    CASE STUDY

To validate that GeoFlow captures meaningful spatial dependencies rather than merely fitting statistical correlations, we visualize the learned attention weights for a representative region, Barbour County, Alabama, United States. We examine the spatial interaction of both global and axial attention mechanism.

In the second panel of Fig. 6, the global attention heatmap shows, for each pair of areas, how much the area in column $j$ contributes to the representation of the area in row $i$. A prominent vertical band

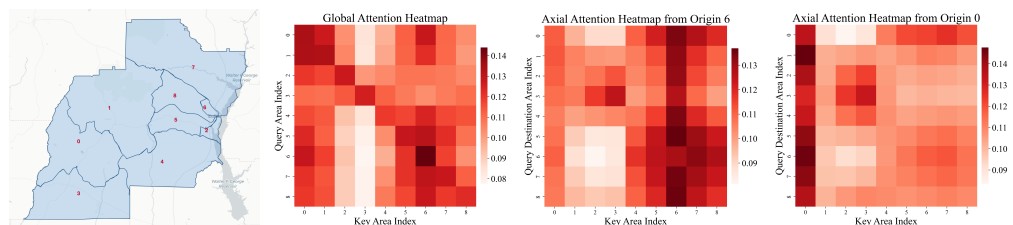

Figure 6: Qualitative case study of learned attention in Barbour County, Alabama, United States. Panels from left to right show the geographic layout of areas, the global attention heatmap, axial attention over destinations for origin area 6, and axial attention over destinations for origin area 0.

of high intensity appears at area 6, which corresponds to Eufaula, the largest city and economic center of Barbour County. This indicates that many other areas place relatively high attention on area 6, consistent with a hub role in the local mobility system. In contrast, area 3, which is located near the county boundary, attracts noticeably less attention from the rest of the county. The weaker column associated with area 3 reflects its more peripheral role and suggests that changes in this edge area have a smaller impact on the representations of other areas.

At a finer scale, the axial attention heatmaps in the third and fourth panels of Fig. 6 show the distribution of attention over destinations for two different origins. The third panel corresponds to an urban center, origin area 6, and exhibits a relatively diffuse pattern with a maximum attention value of about $0.137$ spread across several neighboring destinations such as indices 5, 7 and 8. This suggests that flows originating from the center are influenced by a combination of multiple nearby areas, and the model distributes attention over a broader neighborhood that matches the multifaceted role of the city core. The fourth panel corresponds to a peripheral origin, area 0, and shows a sharper and more self focused pattern, with a maximum attention value of about $0.148$ concentrated on index 0. For this edge area, outflow is therefore governed mainly by the intrinsic characteristics of the origin rather than by strong interactions with alternative destinations. The contrast between diffuse urban attention and focused peripheral attention indicates that GeoFlow reflects the spatial heterogeneity of commuting flows and adjusts its effective receptive field to the functional role of each area, rather than relying only on distance as in classical gravity models.

### F.2 VISUALIZATION

To qualitatively assess the performance of GeoFlow, we randomly select several regions for visualization. As illustrated in Fig. 7, for each region we provide its geographic map, the ground truth OD flow, the model prediction, and three independently generated samples. The prediction results are designed to closely approximate the most likely OD flows, while the generated samples reflect the overall flow patterns while preserving key structural characteristics. This comparison demonstrates that GeoFlow not only accurately predicts dominant flows but also effectively captures the variability and essential features of the underlying OD distribution.

## G DISCUSSION OF BASELINE METHODS

To provide a comprehensive evaluation of GeoFlow, we compare against a representative set of baselines spanning principle-driven, machine learning, and deep learning paradigms. In this section, we summarize their main characteristics, strengths, and limitations.

**Gravity Model.** The gravity model is among the earliest and most widely used approaches to OD flow modeling. It posits that flows are positively associated with the "mass" (e.g., population or economic activity) of the origin and destination and inversely associated with the distance between them, typically through a power-law or exponential decay. Its transparent mathematical form provides strong interpretability and has long served as a benchmark in mobility studies. However, its reliance on simple parametric functions limits flexibility in capturing heterogeneous and nonlinear mobility patterns observed in real data.

**Classical Machine Learning Regressors.** Tree-based models such as Random Forest (RF) and Gradient Boosted Regression Trees (GBRT), as well as kernel-based methods like Support Vector Regression (SVR), represent the next stage in OD flow modeling. These approaches directly learn mappings from urban attributes to flow values. RF and GBRT benefit from their ability to capture nonlinear feature interactions and to provide feature importance measures, while SVR leverages kernel functions to model similarity in high-dimensional feature spaces. Although they improve predictive accuracy over principle-driven models, their reliance on hand-crafted features and limited ability to capture spatial dependencies constrain their performance on large, complex urban systems.

**Deep Learning Models.** The Deep Gravity Model (DGM) builds on the intuition of gravity models but employs neural networks to learn flexible production, attraction, and impedance functions. This extension enables DGM to capture nonlinearities beyond traditional formulations while maintaining interpretability to some degree. The Geographical Multi-task Embedding Learning model (GMEL) incorporates graph attention networks to integrate spatial and structural signals, offering improved representation learning. However, these models generally treat areas as independent entities with limited consideration of long-range spatial interactions, which can restrict their expressiveness.

**Graph-based Generative Models.** Recent advances have explored graph neural networks and generative architectures to model OD flows as attributed graphs. NetGAN employs random walks to learn latent graph structures and can synthesize plausible OD matrices, but struggles with capturing fine-grained attributes. DiffODGen introduces diffusion processes to jointly model topology and edge weights, while WEDAN applies diffusion-based denoising to learn conditional flow generation. These methods capture complex structural dependencies and achieve strong generative performance, yet they typically abstract areas as nodes without explicit treatment of spatial proximity, limiting their ability to represent geographic relations that influence mobility.

**Transformer-based Methods.** TransFlower adapts transformer architectures to OD flow modeling by introducing relative positional embeddings, highlighting the importance of spatial relations between origins and destinations. While it demonstrates strong performance, its reliance on pairwise attention incurs high computational cost and can be difficult to scale to large urban systems. Furthermore, the integration of topological information remains limited, leaving room for improvement in modeling multi-relational geographic contexts.

In summary, principle-driven models emphasize interpretability but are constrained in flexibility; machine learning regressors improve predictive power but fail to capture structural dependencies; deep learning models introduce nonlinear representation learning but often overlook geographic context; and graph generative and transformer-based methods achieve strong expressiveness at the expense of interpretability or efficiency. GeoFlow is designed to address these gaps by explicitly incorporating multi-relational geographic structures while maintaining scalability, enabling more faithful modeling of OD flows across diverse urban contexts.

## H    CLARIFICATION OF LARGE LANGUAGE MODEL USAGE

In this work, we use large language models (LLMs) as auxiliary tools for manuscript polishing and IDE's built-in code completion. All scientific contributions, including algorithmic innovations, experimental design, results, and interpretations, are conceived and produced by authors. Any text or code generated by LLMs is reviewed and edited by authors to ensure technical correctness and fidelity to the intended meaning. The authors accept full responsibility for the submitted contents.

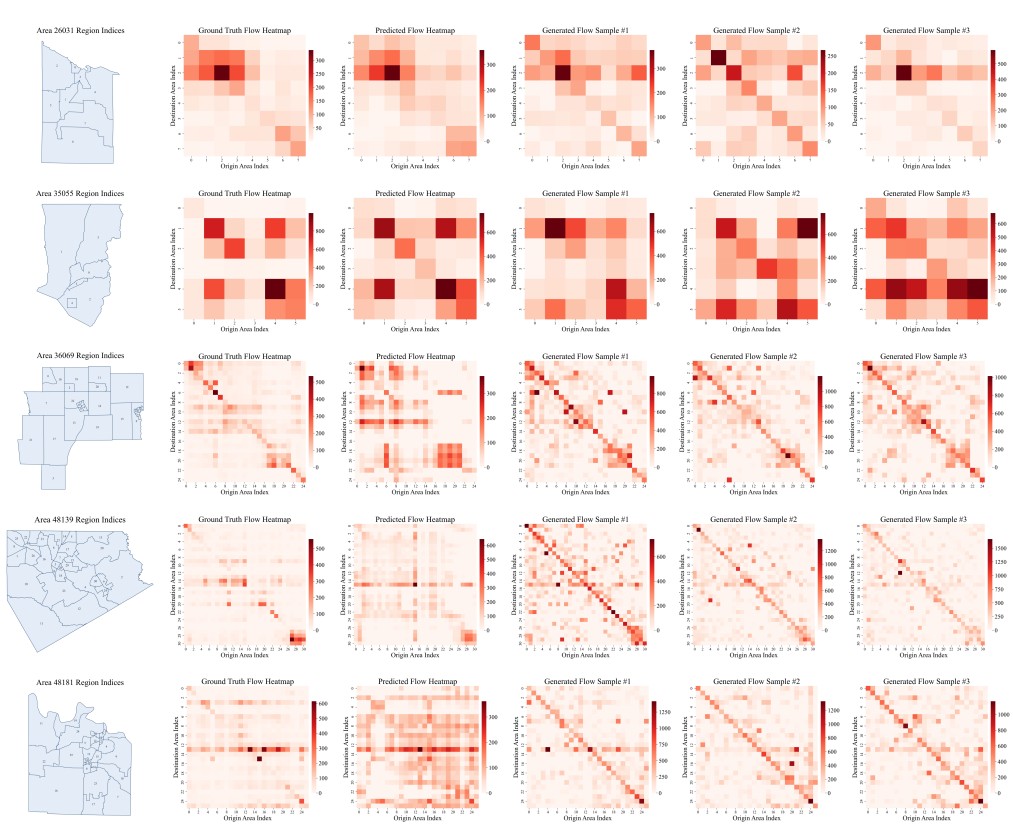

Figure 7: Qualitative comparison of OD flows. For each selected region, we show the geographic map, the ground truth, the model prediction, and three independently generated samples.

