# OpenReview forum: "GeoFlow: Geo-Aware Modeling of Inter-Area Relationships in OD Flow Prediction and Generation"
_ICLR.cc/2026/Conference — Submitted to ICLR 2026_

### Official Review · Reviewer_hQRv · 2025-10-25

**Soundness:** 2
**Presentation:** 3
**Contribution:** 2
**Rating:** 4
**Confidence:** 5

**Summary:**

This paper introduces GeoFlow, a novel framework for origin-destination (OD) flow prediction and generation that systematically integrates geospatial attributes into a unified encoder-decoder architecture. The key contributions include:
(1) Geospatial feature augmentation with relative position, k-hop distance, and geodesic distance to capture long-range and multi-area dependencies.
(2) A geometric-intrinsic fusion encoder that combines graph attention for local attributes with coordinate-aware encoding for global structure.
(3) An axial-global attention decoder that efficiently models competitive dependencies among OD pairs while reducing computational complexity.

Experiments on the CommutingODGen dataset show that GeoFlow achieves state-of-the-art performance in both prediction and generation tasks, with significant improvements in CPC, RMSE, MAE, and diversity metrics.

**Strengths:**

1. The integration of geospatial augmentation, geometric-intrinsic fusion encoding, and axial-global decoding is well-motivated.

2. The paper is exceptionally well-written, with clear explanations of motivations, methods, and results.

3. The model achieves significant performance gains while maintaining computational efficiency, making it suitable for real-world deployment.

**Weaknesses:**

Need more exploration on interpretability.

**Questions:**

1. The model is evaluated only on commuting data. How well does it generalize to other types of OD flows (e.g., tourism, logistics)?

2. While the axial attention reduces cost, what is the total training/inference time compared to baselines like TransFlower?

---

> ### Author Response · Authors · 2025-11-25
> **Response to Reviewer hQRv (part 1)**
>
> We thank you for your careful reading of our paper and for the constructive comments that help us improve the work. Below we respond to each point in turn. All changes in the revised manuscript are marked in blue.
>
> **[W1] More exploration on interpretability**
>
> Thank you for pointing this out. In the revised version we try to better explain not only what GeoFlow achieves, but also how and when it does so, by expanding both qualitative and quantitative analyses.
>
> On the interpretability side, we add a dedicated case study in Appendix F.1 where we visualize the learned global and axial attention in Barbour County, Alabama, United States. The global attention heatmap reveals a clear hub–periphery pattern: the largest city in the county receives strong attention from many other areas, while peripheral areas near the boundary receive much weaker attention. The axial attention maps then contrast an urban origin, whose attention is spread over multiple attractive destinations, with a peripheral origin, whose attention is sharply concentrated on itself. These patterns are consistent with known commuting structure and provide an interpretable view of proposed axial-global attention mechanism.
>
> To complement this, we extend the quantitative analysis of when the model works well or poorly. Beyond the original area-scale generalization study (Appendix C.1.1, Table 6), we add a new geography-transfer experiment in the generation setting (Appendix C.1.2), where we train on eastern, western, or random partitions and evaluate on the held-out subset. We also add two performance profiles in Appendix C.1.3 (Tables 8 and 9) that break down results by flow magnitude and OD distance. Taken together, these experiments show that GeoFlow is most reliable on small and medium regions, on medium and high flows, and on short to medium distances, while very large, heterogeneous regions and extremely low-flow or very long-distance pairs are intrinsically harder. This helps explain the mixed quality in some of the visual examples and links model behaviour to concrete spatial and demand regimes.
>
>
>
> | ID   | Flow scale         | Proportion | Inv. Areas | CPC $\uparrow$ | RMSE $\downarrow$ | NRMSE $\downarrow$ |
> | ---- | ------------------ | ---------- | ---------: | ---------------- | ------------------- | -------------------- |
> | 1    | \(0 \le < 5\)      | 83.18%     |     85.84% | 0.105            | 146.1               | 219.1                |
> | 2    | \(5 \le < 20\)     | 11.62%     |     96.46% | 0.269            | 185.7               | 19.56                |
> | 3    | \(20 \le < 100\)   | 4.49%      |    100.00% | 0.493            | 256.1               | 6.404                |
> | 4    | \(100 \le < 500\)  | 0.69%      |     99.71% | 0.620            | 373.9               | 2.063                |
> | 5    | \(500 \le < 1000\) | 0.02%      |     48.08% | 0.640            | 665.9               | 1.036                |
> | 6    | \(\ge 1000\)       | 0.01%      |      8.55% | 0.513            | 1330                | 1.010                |
>
> | ID   | Distance scale    | Proportion | Inv. Areas | CPC $\uparrow$ | RMSE $\downarrow$ | NRMSE $\downarrow$ |
> | ---- | ----------------- | ---------- | ---------: | ---------------- | ------------------- | -------------------- |
> | 1    | \(0 \le < 0.1\)   | 4.67%      |    100.00% | 0.556            | 447.9               | 22.56                |
> | 2    | \(0.1 \le < 0.2\) | 8.66%      |     49.85% | 0.418            | 389.2               | 46.27                |
> | 3    | \(0.2 \le < 0.5\) | 31.13%     |     73.16% | 0.471            | 244.5               | 47.34                |
> | 4    | \(0.5 \le < 1\)   | 40.33%     |     95.28% | 0.483            | 163.4               | 55.51                |
> | 5    | \(1 \le < 2\)     | 15.16%     |    100.00% | 0.408            | 122.6               | 46.37                |
> | 6    | \(\ge 2\)         | 0.05%      |     39.82% | 0.251            | 64.62               | 8.626                |

---

> ### Author Response · Authors · 2025-11-25
> **Response to Reviewer hQRv (part 2)**
>
> Additionally, we extend the scalability analysis in Appendix C.2. A capacity study (Table 7) varies the hidden dimension and the number of axial–global attention blocks, and we observe consistent but diminishing improvements as capacity increases, indicating that GeoFlow can effectively exploit additional parameters while maintaining stable training behaviour.
>
> | ID   | Hidden Dim. | \# Encoder Attn Layers | \# Decoder Attn Layers | \# Attn Heads | CPC $\uparrow$ | RMSE $\downarrow$ | JSD $\downarrow$ |
> | ---- | ----------- | ---------------------- | ---------------------- | ------------- | ---------------- | ------------------- | ------------------ |
> | 1    | 16          | 1                      | 1                      | 2             | 0.576            | 107.5               | 0.212              |
> | 2    | 32          | 1                      | 1                      | 2             | 0.582            | 98.59               | 0.200              |
> | 3    | 64          | 1                      | 1                      | 2             | 0.593            | 92.43               | 0.194              |
> | 4    | 32          | 2                      | 2                      | 4             | 0.601            | 86.98               | 0.189              |
>
> **[Q1] Performance beyond commuting flows**
>
> We agree that testing on tourism and logistics flows would be very valuable. However, as far as we are aware, there is not yet a widely used, large-scale benchmark for tourism or logistics OD flows with the same level of standardization and quality control as census-based commuting matrices. While some tourism and freight data sources are publicly available, turning them into well-curated benchmarks with consistent preprocessing and reliable ground truth would require substantial additional work. For this reason, we follow prior representative studies, such as DeepGravity [1], TransFlower [2], and WEDAN [3], who focused their empirical evaluation on commuting data, to adhere to established practice.
>
> Within this setting, we choose the CommutingODGen [3] dataset. To the best of our knowledge, it is one of the largest and most recent open datasets for mobility flows and already benchmark SOTA baselines. This allows us to position GeoFlow in a fair and reproducible way. Architecturally, GeoFlow does not encode assumptions specific to commuting: it operates on OD graphs with geographic descriptors and intrinsic attributes, and in principle the same design could be applied to tourism or logistics flows once appropriate OD matrices and covariates are available. However, since we have not yet run such experiments, we prefer not to overstate its generality and instead view this as a natural and important direction for future work. If you are aware of robust tourism or logistics OD benchmarks that would be suitable for this type of study, we would be very grateful for any pointers, and we would be keen to explore them in future extensions of this work.
>
>
> **[Q2] The total training/inference time compared to TransFlower?**
>
> We profile GeoFlow and TransFlower. The table below reports average training and inference wall-clock time per step, together with FLOPs and peak memory, measured after warm-up.
>
> |        | Training Time (s) | Inference Time (s) | Avg. FLOPs             | Peak FLOPs              | Peak Mem. (MiB) |
> |------------|-------------------|--------------------|------------------------|-------------------------|-----------------|
> | TransFlower | 0.0580            | 0.4468             | $1.49 \times 10^{8}$ | $8.82 \times 10^{10}$ | $2.31 \times 10^{4}$ |
> | GeoFlow     | 0.0160            | 0.1104             | $6.12 \times 10^{8}$ | $3.58 \times 10^{11}$ | $2.23 \times 10^{4}$ |
>
> Although GeoFlow has higher average FLOPs per step (about $4\times$ more), its axial–global design parallelizes better on GPU, so training is roughly $3.6\times$ faster and inference about $4\times$ faster than TransFlower, with similar peak memory usage. In other words, the added structure of axial–global attention does not make GeoFlow slower in practice. Instead, it leads to better hardware utilization while improving accuracy. We note that the exact numbers may vary with implementation details and hyperparameter choices. We try to keep the engineering effort and optimization level comparable across models.
>
> Regarding **total** training time to convergence, this naturally depends on the learning schedule and early-stopping criterion. With our default settings and validation-based early stopping, GeoFlow converges on the CommutingODGen benchmark within about four hours on a single general-purpose GPU (e.g., an NVIDIA GeForce RTX 4090) per benchmark configuration during training, which we consider a reasonable cost for city-scale OD modeling. We hope this helps clarify the practical training and inference cost relative to TransFlower and address your concerns.

---

> ### Author Response · Authors · 2025-11-25
> **Response to Reviewer hQRv (part 3)**
>
> We hope the clarifications and new results address your concerns. Should you find them helpful, we would kindly ask you to consider raising your scores accordingly. If any points remain ambiguous or incomplete, please feel free to ask and we will gladly elaborate further.
>
> *Reference*
>
> [1] Simini et al. A Deep Gravity model for mobility flows generation. *Nature Communications*, 12(1):6576, 2021.
>
> [2] Luo et al. TransFlower: An Explainable Transformer-Based Model with Flow-to-Flow Attention for Commuting Flow Prediction. arXiv:2402.15398.
>
> [3] Rong et al. A Large-scale Dataset and Benchmark for Commuting Origin-Destination Flow Generation. ICLR 2025.

---

> > ### Comment · Reviewer_hQRv · 2025-11-26
> >
> > Thanks for the response from the authors. In the revised manuscript, the authors claim that "Table 11 reports average training and inference wall-clock time, average and peak FLOPs, and peak GPU memory usage, computed using the same codebase, dataset, and optimization settings as in the main experiments." Could the authors provide the code implementation of TransFlower using their codebase, to ensure that the efficiency experiments are reproducible?

---

> > > ### Author Response · Authors · 2025-11-26
> > >
> > > Thank you for your prompt response. The sentence “computed using the same codebase, dataset, and optimization settings as in the main experiments” is intended to mean that, TransFlower's efficiency numbers are measured under same configuration as those used to obtain its main performance results reported in the main text. GeoFlow and TransFlower are implemented in different codebases. We will adjust this wording in the next version to reflect this more precisely and avoid misunderstanding. Thank you for pointing this out.
> > >
> > > The TransFlower paper ([link](https://arxiv.org/pdf/2402.15398)) provides a code link that is currently no longer accessible, and, as far as we know, TransFlower remains a preprint rather than a formally published paper. In this context, we are cautious about releasing code under the TransFlower name, so as not to implicitly present our reimplementation as an official version. If you feel an unofficial reimplementation would be helpful for the community, we are open to cleaning up our internal implementation and releasing it later as an explicitly unofficial TransFlower baseline.

---

### Official Review · Reviewer_Tew9 · 2025-11-01

**Soundness:** 3
**Presentation:** 2
**Contribution:** 3
**Rating:** 4
**Confidence:** 4

**Summary:**

This paper presents GeoFlow, a new method for origin-destination flow prediction and generation. The main novelty is 3 folds: 1) GeoFlows uses four different geospatial relationship features to jointly capture the origin-destination relation; 2) The Axial-global attention is proposed to perform message passing; 3) The GeoFlow can perform both prediction and generation tasks.

**Strengths:**

1. The experiments clearly show the advantages of GeoFlow over other methods;
2. A set of ablation studies are conducted to test each component of this model.

**Weaknesses:**

1. Theretically speaking, the relative position and the straight-line distance have the same information. Why use both in the framework? Another ablation setting is needed, which drops the straight-line distance and only uses the other 3 features for flow prediction.
2. Compared with the baselines, GeoFlow performs Axial-global attentions, which lead to higher computational complexity. Please compare its computational complexity with baselines.
3. Equation 6 and 7 are hard to understand. What do Z^(2k) and Z^(2k+1) mean? The formulas lack definitions, which makes this paper hard to understand.
4. What is the difference between A_{axial} in Equation 4 and A^(L_(a))_{axial} in Equation 8?

**Questions:**

See above.

---

> ### Author Response · Authors · 2025-11-25
> **Response to Reviewer Tew9 (part 1)**
>
> We thank you for the thoughtful comments. We address each point below, and all corresponding changes in the manuscript are highlighted in blue.
>
> **[W1] The necessity of straight-line distance**
>
> We agree that, in principle, normalized relative position contain essential information for deriving straight-line (SL) distance, and a sufficiently expressive model can learn the simple mapping between them. We keep both mainly to follow the standard practice in previous methods, where straight-line distance is usually included as the most direct and interpretable distance signal. To address your suggestion, we conduct an ablation where we remove the straight-line distance and keep only the remaining three geometric features.
>
> | Model                | CPC $\uparrow$ | RMSE $\downarrow$ | JSD $\downarrow$ |
> | -------------------- | -------------- | ----------------- | ---------------- |
> | GeoFlow w/o SL feat. | 0.601          | 86.68             | 0.172            |
> | GeoFlow (full geom.) | 0.604          | 85.64             | 0.171            |
>
> The differences are very minor, which confirms that the model can indeed recover the relevant distance information from relative positions alone and that our results are not sensitive to including straight-line distance. Dropping this feature slightly reduces the input dimensionality and parameter count, but the savings are negligible compared to the total model size. For clarity and comparability with prior work, we therefore keep straight-line distance in the main configuration.
>
> **[W2] Computational complexity compared with baselines**
>
> From a theoretical perspective, it is difficult to give a fully fair complexity comparison across all architectures, because different methods change the computation graph in heterogeneous ways. For example, DiffODGen [1] first trains a separate neural network to decide whether an OD pair has nonzero flow, which avoids full attention in theory but introduces an additional classifier and screening step in practice. Similarly, diffusion-based generators require multiple sampling steps whose cost depends on the chosen schedule and target quality. In such settings, purely asymptotic big-O comparisons are often not very informative.
>
> For this reason, we focus on **empirical** complexity under a controlled setup and report standard hardware-aware metrics. In Appendix D Table 11, we profile GeoFlow against the transformer-based prediction baseline TransFlower [4]. We report training and inference wall-clock time per case, average and peak FLOPs, and peak GPU memory. GeoFlow has higher average FLOPs than TransFlower (approximately $6.1\times10^8$ vs. $1.5\times10^8$), but due to its more parallel axial–global design it achieves shorter wall-clock time per case. Training is about $3.6\times$ faster and inference about $4\times$ faster, while peak memory remains at a similar level for both models. This suggests that, in practice, GeoFlow attains a more favorable accuracy–efficiency trade-off on GPU hardware, even though its per-step FLOP count is larger. For the generation baselines, the SOTA methods follow diffusion-based designs (e.g., WEDAN [5]), whose total cost scales with the number of sampling steps. Recent studies [2, 3] show that flow matching can reach comparable or better quality with substantially fewer function evaluations than diffusion.
>
> |             | Training Time (s) | Inference Time (s) | Avg. FLOPs             | Peak FLOPs              | Peak Mem. (MiB)        |
> | ----------- | ----------------- | ------------------ | ---------------------- | ----------------------- | ---------------------- |
> | TransFlower | 0.0580            | 0.4468             | $1.49 \times 10^{8}$ | $8.82 \times 10^{10}$ | $2.31 \times 10^{4}$ |
> | GeoFlow     | 0.0160            | 0.1104             | $6.12 \times 10^{8}$ | $3.58 \times 10^{11}$ | $2.23 \times 10^{4}$ |
>
> We clarify this discussion in the appendix and add the empirical profiling table to make the comparison more transparent.
>
>
>
> *Reference*
>
> [1] Rong et al. Complexity-aware Large Scale Origin-Destination Network Generation via Diffusion Model. arXiv 2306.04873.
>
> [2] Schusterbauer et al. Diff2Flow: Training Flow Matching Models via Diffusion Model Alignment. CVPR 2025.
>
> [3] Gui et al. DepthFM: Fast Generative Monocular Depth Estimation with Flow Matching. AAAI 2025.
>
> [4] Luo et al. TransFlower: An Explainable Transformer-Based Model with Flow-to-Flow Attention for Commuting Flow Prediction. arXiv:2402.15398.
>
> [5] Rong et al. A Large-scale Dataset and Benchmark for Commuting Origin-Destination Flow Generation. ICLR 2025.

---

> ### Author Response · Authors · 2025-11-25
> **Response to Reviewer Tew9 (part 2)**
>
> **[W3] Notation and meanings of $Z^{(2k)}$ and $Z^{(2k+1)}$**
>
> Thank you for pointing out that the notation in the decoder was not sufficiently self-contained. We agree this made the forward pass harder to follow.
>
> In the revised version, we describe the decoder more sequentially and define all intermediate tensors locally in the decoder section. We state that the decoder takes the encoder output $Z$ from Eq. (5) as input and initialize $Z^{(0)} = Z$. We then explain that each axial layer consists of two sublayers: an origin-wise attention sublayer followed by a destination-wise attention sublayer. For the $(k+1)$-th axial layer, $Z^{(2k)}$ is the tensor fed into the origin-wise sublayer, whose output is $Z^{(2k+1)}$. It then pass to the destination-wise sublayer, whose output is $Z^{(2k+2)}$. After stacking $L_a$ such layers, we obtain a sequence $Z^{(0)}, Z^{(1)}, \dots, Z^{(2L_a)}$, where $Z^{(2L_a)}$ is explicitly defined as the output of the axial attention stack.
>
> These definitions are now given immediately before the decoder equations and we avoid referring to notation introduced elsewhere, so that the decoder block can be read and understood on its own. We hope this makes the meaning of the notation clearer.
>
> **[W4] Difference between $\mathcal{A}$\_${\mathrm{axial}}$ in Eq. (4) and $\mathcal{A}^{L_a}_{\mathrm{axial}}$ in Eq. (8)**
>
> Thank you for asking for clarification on this notation. In the revised version we make the distinction more explicit. Conceptually, Eq. (4) uses an **encoder-side** axial attention operator that takes scores derived from the geographic tensor $G$ and is used once to aggregate intrinsic area features into $\widetilde{E_x}$. Eq. (8) instead denotes the **decoder-side** axial attention stack that operates on the OD embedding tensor $Z$ and is applied $L_a$ times in sequence.
>
> To avoid confusion, we now explicitly refer to the encoder operator as $\mathcal{A}^{\mathrm{enc}}$\_${\mathrm{axial}}$, which is instantiated on $G$ with its own parameters, and to the decoder stack as $\mathcal{A}^{L_a}_{\mathrm{axial}}$, which is the composition of $L_a$ origin-wise and destination-wise attention layers acting on $Z$. Both share the same functional form, but they are applied at different stages, to different inputs, and with separate parameter sets. We hope this clarifies the relationship between the two notations.
>
>
> We hope the response clarifies our approach and resolves the issues you raised. We would be thankful if you could consider updating your scores if the concerns have been well addressed. We are happy to provide any additional clarification if needed.

---

### Official Review · Reviewer_3QvF · 2025-11-01

**Soundness:** 3
**Presentation:** 2
**Contribution:** 3
**Rating:** 4
**Confidence:** 4

**Summary:**

The paper proposed GeoFlow model for origin-destination flow map prediction and generation. The model integrates various geo-spatial information, intrinsic local information, and graph information via multiple attention mechanisms and finally produces a set of informative embeddings for each OD pair. These embeddings are then used for supervised OD-flow prediction as inputs and OD-flow generation via a flow-matching model as conditions. The predicted OD-flows provides the best flow value estimations, and the generated samples present possible realizations according to the underlying distribution dynamics. The method achieve the new SOTA of the related tasks. Systematical ablation studies and analysis are conducted to explore the effectiveness of the main model components. A few issues exist in the paper, including model architecture, computational complexity, baseline model implementation, and explanability.  I may change my score if the concerns are properly addressed.

**Strengths:**

1. The work develops a novel geo-spatial and local feature integration methodogy based on multiple hierachical attention mechanisms.
2. It achieves the SOTA performance not only in supervised OD-flow prediction but also in unsupervised flow map generation tasks. For the latter, this works presents the first flow-matching model for OD-flow map generation, which holds good performance, training stability, and sample diversity.
3. Ablation studies reveal the importance of k-hop distance and geodesic distance, which possess unique information that cannot be represented by coordinates.

**Weaknesses:**

1. The model architecture. The final OD pair embedding Z is given by Eq. (5), which depends on the area embedding given by Eq.(4). These area embeddings defined in Eq. (4) depends on two types of attentions, A_axial and A_global. However, the problem is A_axial and A_global further depend the final embedding Z, as specified in Eqs. (6), (7), and (9), and that makes the forward propagation a circle. Do the authors actually made a typo that the Z in Eqs. (6), (7), and (9) should be the geographic tensor G? Please clarify.
2. Computational complexity. The authors claim to have lower computational complexity than O(N^4). For A_axial, I understand it can be reduced to O(N^3). But for A_global, it is not clearly explained how to make it smaller than O(N^4). Please provide better explanation.
3. The Transflower baseline. Transflower only uses local features and coordinates without the information of k-hop distance and geodesic distance, Is the performance advantage of GeoFlow mainly because it uses more information? In Table2, does ID 2 corresponds to the same amount of information used by Transflower? If yes, then the model actually doesn't show advantage. If not, please provide this ablation. It is important to distinguish the gains from better model design and from more information.
4. Explanability of attention map. The attention map of Transflower has clear correlation to the points of interests. What about A_axial and A_global of GeoFlow?
5. Explanability to performances. In Fig. 6 in appendix, the first two rows show decent similarity between the ground truth and both the prediction and the generation samples. But the last two rows seem to show the prediction and generation are poor. Do we know in what case (what geo or intrinsic properties of the areas) the method can or can not work well?

**Questions:**

See weaknesses.

---

> ### Author Response · Authors · 2025-11-25
> **Response to Reviewer 3QvF (part 2)**
>
> **[W3] TransFlower baseline with additional information**
>
> GeoFlow does benefit from additional geometric features, but its advantage is not limited to that. There are two core architectural differences relative to TransFlower:
>
> 1. **Encoder for geometric–intrinsic fusion.** GeoFlow uses an attention-based encoder that fuses geometric attributes and intrinsic area attributes into a unified OD representation. In contrast, TransFlower processes local features and coordinates more shallowly, through separate encoders that are concatenated later, with less structured interaction between geometric and intrinsic signals.
> 2. **Axial–global attention over OD pairs.** GeoFlow introduces an axial–global attention mechanism that explicitly models competition and allocation among OD pairs and enables region-level information exchange even when the number of areas is large. TransFlower focuses on pair-wise interactions at the OD level and does not include this explicit region-level aggregation and broadcasting.
>
> These architectural components are ablated in the main paper (Table 3 and Figure 3) and show substantial contributions beyond feature choice.
>
> Regarding Table 2, ID 2 is not intended to be a reimplementation of TransFlower with exactly the same information and architecture. It is an internal ablation within the GeoFlow family. To directly address the fairness concern, we conduct a dedicated baseline where we extend TransFlower with the same additional geometric features (k-hop distance and geodesic distance) used by GeoFlow, while keeping the rest of TransFlower unchanged.
>
> |                                  | CPC $\uparrow$ | RMSE $\downarrow$ | JSD $\downarrow$ |
> | -------------------------------- | -------------- | ----------------- | ---------------- |
> | TransFlower                      | 0.486          | 106.4             | 0.281            |
> | TransFlower w/ extra geom. feat. | 0.492          | 104.3             | 0.269            |
> | GeoFlow$_\text{prediction}$      | **0.604**      | **85.64**         | **0.171**        |
>
> Adding k-hop and geodesic distance gives TransFlower a small improvement, which confirms that richer geometric information is helpful on its own. However, even with these extra features, GeoFlow remains substantially better wrt. metrics. This indicates that the gains contributed from the architectural design of GeoFlow, with the additional features providing a complementary boost rather than being the sole source of improvement. We hope this clarifies the distinction between gains due to more information and gains due to the proposed model design.
>
> **[W4] Explainability of attention maps for $A_{\text{axial}}$ and $A_{\text{global}}$**
>
> In the revised version, we include an qualitative case study for GeoFlow in Appendix F.1. We visualize both the global attention and the axial attention for a representative county, Barbour County, Alabama, United States (Figure 10). The global attention heatmap shows, for each pair of areas, how much the area in column $j$ contributes to the representation of the area in row $i$. In this view, area 6, which corresponds to Eufaula, the largest city and economic center of the county, forms a clear vertical band with high attention from many other areas. By contrast, area 3, located near the county boundary, receives noticeably weaker attention. This pattern matches standard monocentric commuting structure and is consistent with the idea that flows in peripheral areas are strongly influenced by the central urban hub, while changes in edge areas have more limited impact on the rest of the region.
>
> At the OD level, we visualize axial attention as heatmaps over destinations for different origins. For an urban origin (area 6), attention is relatively diffuse, with mass spread over several nearby destinations, reflecting the fact that trips from the center are allocated across multiple attractive zones. For a peripheral origin (area 0), attention is much more concentrated on the origin itself, suggesting that outflows there are governed mainly by local characteristics and that alternative destinations play a smaller role. Taken together, these global and axial patterns provide an interpretable picture of how GeoFlow encodes spatial interactions, in a way that aligns with known urban structure and complements the POI-based explanation given for TransFlower.

---

> ### Author Response · Authors · 2025-11-25
> **Response to Reviewer 3QvF (part 3)**
>
> **[W5] When does the method work well or poorly?**
>
> In Figure 6 (Figure 7 in the revised numbering) the examples were randomly sampled rather than cherry-picked, so it is expected that some panels will show difficult cases. We agree that this may give a somewhat pessimistic impression if a few hard examples dominate the figure. In a later revised version we will replace these with more representative and better annotated case studies to avoid potential case-selection bias and to provide clearer qualitative insight.
>
> Beyond individual visual examples, we use the analysis experiments (Table 6 and the new Tables 8 and 9 in the appendix) to answer your question more systematically. Table 6 shows that performance depends strongly on **area scale**: small and medium regions are both numerically dominant and structurally simpler, and GeoFlow achieves the best CPC and lowest error there, while large regions are fewer and much more heterogeneous, and accordingly they are the hardest to model. Tables 8 and 9 further break down performance in the generation setting by **flow magnitude** and **OD distance**. The model is most reliable for medium and high flows and for short to medium distances, which correspond to the bulk of real trips; low-flow and very long-distance pairs are both extremely sparse and noisy, so even small absolute deviations can translate into low CPC and visually poorer samples.
>
> | ID   | Flow scale         | Proportion | Inv. Areas | CPC $\uparrow$ | RMSE $\downarrow$ | NRMSE $\downarrow$ |
> | ---- | ------------------ | ---------- | ---------: | ---------------- | ------------------- | -------------------- |
> | 1    | \(0 \le < 5\)      | 83.18%     |     85.84% | 0.105            | 146.1               | 219.1                |
> | 2    | \(5 \le < 20\)     | 11.62%     |     96.46% | 0.269            | 185.7               | 19.56                |
> | 3    | \(20 \le < 100\)   | 4.49%      |    100.00% | 0.493            | 256.1               | 6.404                |
> | 4    | \(100 \le < 500\)  | 0.69%      |     99.71% | 0.620            | 373.9               | 2.063                |
> | 5    | \(500 \le < 1000\) | 0.02%      |     48.08% | 0.640            | 665.9               | 1.036                |
> | 6    | \(\ge 1000\)       | 0.01%      |      8.55% | 0.513            | 1330                | 1.010                |
>
> | ID   | Distance scale    | Proportion | Inv. Areas | CPC $\uparrow$ | RMSE $\downarrow$ | NRMSE $\downarrow$ |
> | ---- | ----------------- | ---------- | ---------: | ---------------- | ------------------- | -------------------- |
> | 1    | \(0 \le < 0.1\)   | 4.67%      |    100.00% | 0.556            | 447.9               | 22.56                |
> | 2    | \(0.1 \le < 0.2\) | 8.66%      |     49.85% | 0.418            | 389.2               | 46.27                |
> | 3    | \(0.2 \le < 0.5\) | 31.13%     |     73.16% | 0.471            | 244.5               | 47.34                |
> | 4    | \(0.5 \le < 1\)   | 40.33%     |     95.28% | 0.483            | 163.4               | 55.51                |
> | 5    | \(1 \le < 2\)     | 15.16%     |    100.00% | 0.408            | 122.6               | 46.37                |
> | 6    | \(\ge 2\)         | 0.05%      |     39.82% | 0.251            | 64.62               | 8.626                |
>
> To sum up, these results suggest that GeoFlow works best for small and medium regions with structured commuting patterns, and for OD pairs that carry nontrivial flow at typical urban and suburban distances. It is least reliable for very large, highly heterogeneous regions and for OD pairs with extremely low flow or very long distances, where the data themselves are scarce and noisy. The dataset is defined at census-area granularity and comes with a large number of potential covariates (points of interest and other geospatial features), so a full attribution from performance to every fine-grained covariate would require a substantial, dedicated study. In this work we focus on the most OD-related dimensions (area scale, flow magnitude, and OD distance) and provide a detailed quantitative analysis along these axes. We view a more comprehensive link to rich POI-level attributes as an interesting direction for future work.
>
> We hope these clarifications and revisions address your concerns, and if they do, we would greatly appreciate it if you could consider reflecting this in your score. Please let us know if anything is still unclear.

---

> ### Author Response · Authors · 2025-11-25
> **Response to Reviewer 3QvF (part 1)**
>
> We thank you for the careful and constructive feedback, which has helped us improve the clarity and completeness of the paper. Below we address each comment point by point, with all revisions marked in blue in the updated manuscript.
>
> **[W1] Clarify circular dependency between area embeddings and attention modules**
>
> Thank you for carefully checking the architecture equations and raising this concern. The use of $Z$ in Eqs. (6), (7), and (9) is intentional and does not create a circular dependency.
>
> In the revised manuscript we clarify the computation order and the roles of the attention modules. The encoder first takes the geographic tensor $G$ and the intrinsic attributes $X$ as input. It uses an encoder-side axial–global attention module, now denoted $\mathcal{A}^{\mathrm{enc}}$\_${\mathrm{axial}}$ (this formula is somehow misrendered) and $\mathcal{A}^{\mathrm{enc}}_{\mathrm{global}}$, which operate on scores derived from $G$ to aggregate intrinsic features and produce $\widetilde{E_x}$ (Eq. (4)). The geographic embedding $E_g$ and the intrinsic embedding $\widetilde{E_x}$ are then fused into the OD representation $Z$ via Eq. (5). This encoder stage depends only on $G$ and $X$ and is fully defined before the decoder is applied.
>
> The decoder then takes $Z$ from Eq. (5) as its input and refines it using a stack of axial and global attention layers, defined in Eqs. (6)–(9). These decoder-side attention modules operate on $Z$ (and its intermediate versions $Z^{(m)})$, have their own parameters, and do not send any information back to the encoder. We also explicitly note in the text that the axial operator is defined in a **generic** way and is instantiated once on the encoder side (with $G$) and once on the decoder side (with $Z$), with independent parameter sets. Under this design, the overall forward pass is strictly feedforward.
>
> We have reorganized the encoder and decoder descriptions to make this flow clearer and to avoid the impression of a circular dependency. We hope this clarification addresses your concern.
>
> **[W2] Computational complexity of the global attention**
>
> Thank you for pointing out that the complexity of the global attention was not clearly explained. We apologise for the confusion caused by the original wording.
>
> In the revised version we make explicit that we never apply full self-attention over all $N^2$ OD pairs. Instead, the global attention first aggregates the OD tensor $Z \in \mathbb{R}^{N \times N \times d_z}$ into $N$ area-level embeddings by averaging over origins and destinations. We then apply standard self-attention to the resulting sequence of length $N$, which costs $O(N^2 d_z)$ for computing the attention scores and the weighted sums. Finally, the attended area-level features are broadcast back to the $N \times N$ OD grid by simple expansions and outer product, which again is $O(N^2 d_z)$, and feedforward to subsequent layers. As a result, the global attention pathway scales as $O(N^2)$ wrt. the number of areas, rather than $O(N^4)$ as in naive full attention over all OD pairs.
>
> Putting this together with the axial attention, which operates along origin and destination axes and has $O(N^3)$ complexity, the overall attention cost of GeoFlow is $O(N^3)$, strictly below the $O(N^4)$ cost of full OD-wise attention (in our empirical profiling, a naive $O(N^4)$ implementation of full OD-wise attention would require more than 1TB of GPU memory for regions with more than 500 areas, which is practically infeasible). We have revised the method section to spell out these steps and their complexity more clearly, and we hope this resolves the concern about the computational cost of $A_{\mathrm{global}}$.

---

### Official Review · Reviewer_u6js · 2025-11-04

**Soundness:** 3
**Presentation:** 3
**Contribution:** 3
**Rating:** 6
**Confidence:** 4

**Summary:**

The paper introduces GeoFlow, a unified framework for flow prediction and generation. GeoFlow integrates geospatial awareness into machine learning models. In particular: (i) it augments area representations with geospatial attributes (e.g., relative positions), (ii) it uses a special encoder module designed to process and combine intrinsic and geometric features, (iii) it relies on an axial–global attention decoder that let the model to understand how trips from the same origin or to the same destination influence each other. GeoFlow is evaluated on existing OD benchmarks and tested against the main SOTA models. GeoFlow outperforms in prediction accuracy over baselines and shows some gain (7.4% relative) in reconstruction accuracy for generation.

**Strengths:**

The paper presents several novelties respect to previous works. For example, the fusion of geometric and intrinsic attributes with flow matching is new in OD modeling. Results are also convincing and the SOTA baselines are take into consideration for the experimental setup.

**Weaknesses:**

While the paper is in-depth and clear for prediction, it lacks of analysis for the generation part. There is a modest gain and I would like to see some inspection or better analysis that justify the improvement (even though is statistically significant, it is not clear where this improvement is).

Scalability is underexplored as well as interpretability and generalization. For example, the geography transferability is not inspected, leaving doubts on the generalization abilities of the model (authors can refer to DeepGravity paper to see examples).

**Questions:**

see weaknesses

---

> ### Author Response · Authors · 2025-11-25
> **Response to Reviewer u6js (part 1)**
>
> We thank you for the careful reading and constructive comments. Below we respond to each point in turn. All changes in the revised manuscript are marked in blue.
>
> **[W1] Analysis of the generation part**
>
> Thank you for raising this point. The improvements in the generation setting come from two parts that we now explain more explicitly. First, the GeoFlow framework provides a shared spatial backbone for both prediction and generation: it learns a latent OD interaction structure that is reused across the two tasks. The contribution of this backbone is supported by ablations in the main paper (Table 2, Table 3, Figure 3), which show that the proposed feature augmentation and encoder–decoder attention over areas are responsible for most of the gains. Second, on top of this backbone we use a flow matching model for the generation tasks. In the appendix we provide an explicit comparison against diffusion models (Figure 5), showing that flow matching leads to more stable training and better performance.
>
> To directly address your request for a more fine-grained analysis of the generation setting, we add a new performance profile in Appendix C.1.3. The first new table (Table 8) reports CPC, RMSE, and normalized RMSE across flow magnitude bins. We observe that more than 80% of OD pairs have very small flows (below 5), where the data are noisy and a few misallocated trips can strongly affect CPC, leading to low apparent correlation. In contrast, for medium and high flows (for example, 20–100 and 100–500), which dominate aggregate mobility, the model achieves much higher CPC (around 0.5–0.6) and substantially lower normalized error.
>
> | ID   | Flow scale         | Proportion | Inv. Areas | CPC $\uparrow$ | RMSE $\downarrow$ | NRMSE $\downarrow$ |
> | ---- | ------------------ | ---------- | ---------: | ---------------- | ------------------- | -------------------- |
> | 1    | \(0 \le < 5\)      | 83.18%     |     85.84% | 0.105            | 146.1               | 219.1                |
> | 2    | \(5 \le < 20\)     | 11.62%     |     96.46% | 0.269            | 185.7               | 19.56                |
> | 3    | \(20 \le < 100\)   | 4.49%      |    100.00% | 0.493            | 256.1               | 6.404                |
> | 4    | \(100 \le < 500\)  | 0.69%      |     99.71% | 0.620            | 373.9               | 2.063                |
> | 5    | \(500 \le < 1000\) | 0.02%      |     48.08% | 0.640            | 665.9               | 1.036                |
> | 6    | \(\ge 1000\)       | 0.01%      |      8.55% | 0.513            | 1330                | 1.010                |
>
> A second table (Table 9) reports the same metrics across normalized OD distance. Most pairs fall into short and medium ranges, where CPC is consistently high and stable, indicating that the model captures local and intra-region patterns well. Very long-distance pairs are extremely sparse, so their CPC is lower, but they account for only a tiny fraction of OD pairs. Besides, we also note that the area-scale performance profile in Appendix C.1.1, although conducted in the prediction setting, may offer complementary intuition for the generation setting, as both rely on the same GeoFlow backbone.
>
> | ID   | Distance scale    | Proportion | Inv. Areas | CPC $\uparrow$ | RMSE $\downarrow$ | NRMSE $\downarrow$ |
> | ---- | ----------------- | ---------- | ---------: | ---------------- | ------------------- | -------------------- |
> | 1    | \(0 \le < 0.1\)   | 4.67%      |    100.00% | 0.556            | 447.9               | 22.56                |
> | 2    | \(0.1 \le < 0.2\) | 8.66%      |     49.85% | 0.418            | 389.2               | 46.27                |
> | 3    | \(0.2 \le < 0.5\) | 31.13%     |     73.16% | 0.471            | 244.5               | 47.34                |
> | 4    | \(0.5 \le < 1\)   | 40.33%     |     95.28% | 0.483            | 163.4               | 55.51                |
> | 5    | \(1 \le < 2\)     | 15.16%     |    100.00% | 0.408            | 122.6               | 46.37                |
> | 6    | \(\ge 2\)         | 0.05%      |     39.82% | 0.251            | 64.62               | 8.626                |
>
> Overall, these additional analyses show that the generative gains are not uniform but are concentrated on high-volume and common-distance OD pairs that matter most in practice, while low-flow and extreme-distance regimes remain intrinsically noisy. We have updated the appendix accordingly, and we hope this clarifies where the improvements in the generation setting come from.

---

> ### Author Response · Authors · 2025-11-25
> **Response to Reviewer u6js (part 2)**
>
> **[W2] Scalability, generalization, and interpretability**
>
> Thank you for pointing out that scalability, geography transferability, and interpretability were not sufficiently explored in the original version. We have revised the paper and appendix to make these aspects more explicit.
>
> For **scalability**, we add a dedicated experiment in the appendix (Table 10, Appendix C.2). We vary model capacity along width (hidden dimension) and depth (number of encoder/decoder attention layers and heads), while keeping all other hyperparameters and training settings fixed. As capacity increases, CPC improves from 0.576 to 0.601, and RMSE and JSD consistently decrease, indicating that GeoFlow can effectively use additional parameters without instability. The gains become smaller as we increase capacity, which is consistent with a regime where performance becomes more data-limited. Further doubling all dimensions leads to out-of-memory on a single GPU, so we focus on the practically relevant range where the model remains deployable. Overall, these results suggest that GeoFlow scales in a controlled and monotonic way within the feasible single-GPU regime.
>
> | ID   | Hidden Dim. | \# Encoder Attn Layers | \# Decoder Attn Layers | \# Attn Heads | CPC $\uparrow$ | RMSE $\downarrow$ | JSD $\downarrow$ |
> | ---- | ----------- | ---------------------- | ---------------------- | ------------- | ---------------- | ------------------- | ------------------ |
> | 1    | 16          | 1                      | 1                      | 2             | 0.576            | 107.5               | 0.212              |
> | 2    | 32          | 1                      | 1                      | 2             | 0.582            | 98.59               | 0.200              |
> | 3    | 64          | 1                      | 1                      | 2             | 0.593            | 92.43               | 0.194              |
> | 4    | 32          | 2                      | 2                      | 4             | 0.601            | 86.98               | 0.189              |
>
> For **geography transferability**, we now report two complementary sets of results. **First**, as discussed in Appendix C.1.1, we study generalization across area scales in the prediction setting (Table 6), where we systematically leave out small, medium, or large areas during training and evaluate cross-scale transfer. This experiment shows that GeoFlow maintains reasonable performance even when an entire scale is omitted, and that including scarce large areas in training improves robustness across all scales. **Second**, to more closely match your suggestion and the spirit of the leave-one-city-out protocol in DeepGravity [1], we add a new experiment in the generation setting (Appendix C.1.2, Table 7). We partition the study region into eastern and western subsets based on longitude and also construct a random held-out partition of the same size. For each configuration, we train on the complement and evaluate on the held-out subset. CPC remains around 0.56–0.57 and JSD varies slightly across these splits, while RMSE differences are moderate. This indicates that GeoFlow does not overfit to a particular longitudinal band and that its generative patterns transfer across distinct geographic subdomains.
>
> | ID   | Train on     | Evaluate on | CPC $\uparrow$ | RMSE $\downarrow$ | JSD $\downarrow$ |
> | ---- | ------------ | ----------- | ---------------- | ------------------- | ------------------ |
> | 1    | All – West   | West        | 0.572            | 111.1               | 0.176              |
> | 2    | All – East   | East        | 0.561            | 116.7               | 0.177              |
> | 3    | All – Random | Random      | 0.566            | 101.2               | 0.174              |
>
> For **interpretability**, we follow the recommendation of TransFlower [2] and add a qualitative case study in Appendix F.1. We visualize both the global attention and the axial attention for a representative county in the United States. The global attention heatmap highlights the largest city in the county as a hub area that receives strong attention from many other areas, while more peripheral areas receive much weaker attention, which aligns with the known monocentric structure of commuting flows there. The axial attention patterns further distinguish between an urban origin, where attention is spread across multiple destinations, and a peripheral origin, where attention is sharply concentrated on the origin itself. These patterns are consistent with domain knowledge about urban cores and edge areas and provide an interpretable view of how GeoFlow encodes spatial interactions.
>
> We have integrated these new experiments and analyses into the revised manuscript. We hope this clarifies the scalability, geography transferability, and interpretability of the proposed model.

---

> ### Author Response · Authors · 2025-11-25
> **Response to Reviewer u6js (part 3)**
>
> We hope these clarifications help address your concerns, and if you find them satisfactory we would be grateful if you could consider updating your score. Feel free to let us know if any questions remain.
>
>
>
> *Reference*
>
> [1] Simini et al. A Deep Gravity model for mobility flows generation. *Nature Communications*, 12(1):6576, 2021.
>
> [2] Luo et al. TransFlower: An Explainable Transformer-Based Model with Flow-to-Flow Attention for Commuting Flow Prediction. arXiv:2402.15398.

---

### Author Response · Authors · 2025-11-25
**General response to all reviewers**

We sincerely thank all reviewers for your careful reading and constructive feedback. In the revised manuscript, we have made the following main changes, all of which are highlighted in blue:

- **Architecture clarification.** We reorganize the encoder–decoder description and add more detailed explanation to avoid any misunderstanding (Section 3.2.2 and 3.2.3).
- **Generalization and transferability.** Beyond the original area-scale generalization study (Appendix C.1.1), we add a new longitudinal partition experiment (Appendix C.1.2) to probe geographic transferability in the spirit of leave-one-city-out evaluation.
- **Performance analysis.** We add a new performance profile for the generation setting, including breakdowns by flow magnitude and OD distance (Appendix C.1.3).

- **Scalability.** We conduct a model scalability study (Appendix C.2) that varies hidden dimension and the number of axial–global attention blocks, showing stable and monotone scaling behaviour.
- **Computational complexity.** We add an empirical profiling comparing training/inference time, FLOPs, and peak memory of GeoFlow and TransFlower (Appendix C.3).
- **Interpretability and case study.** We add a qualitative case study of global and axial attention in Barbour County, Alabama, USA (Appendix F.1), and connect it to the quantitative performance profiles to enhance interpretability.

Due to the tight rebuttal timeline, there may still be minor issues in formatting and stylistic consistency. We will continue to polish these in later revisions. If anything remains unclear or if further details would be helpful, please feel free to ask.

---

> ### Author Response · Authors · 2025-12-03
>
> The new revised manuscript has been uploaded. In this version, we make the following changes:
>
> 1. In response to Reviewer 3QvF, we update the visualization in Figure 7 to avoid potential bias from case selection. We also adjust its color scheme and style to be consistent with Figure 6 for better visual coherence.
> 2. In response to Reviewer hQRv, we revise the description of the TransFlower baseline configurations in Appendix D to avoid possible misunderstandings.
>
> We hope these changes further improve the clarity and quality.

---

### Author Response · Authors · 2025-12-03
**Rebuttal and Revision Summary**

We thank the reviewers and the Area Chair for their constructive feedback and valuable work. We summarize below the concerns raised by reviewers and our rebuttal and revision.

**Method summary:** GeoFlow is a unified framework for origin-destination flow prediction and generation that systematically incorporates geographic relationships via (i) geospatial feature augmentation (relative position, k-hop, geodesic distance), (ii) a geometric–intrinsic fusion encoder, and (iii) an axial–global attention decoder. For generation, it is paired with a flow matching model to improve realism and diversity.

**Key concerns raised by reviewers:**

- **Architecture correctness and presentation** (Reviewer 3QvF, Tew9)

    We clarified the computation order and notation and reorganized corresponding sections.

- **Complexity of global attention** (Reviewer 3QvF)

    We explained that global attention operates on an area-level sequence, not all $N^2$ OD pairs, and provided empirical profiling.

- **Fairness vs. TransFlower** (Reviewer 3QvF)

    We added a “TransFlower + same extra geometric features” baseline, showing that GeoFlow’s advantage mainly comes from the architecture rather than additional inputs.

- **Performance analysis** (Reviewer u6js, 3QvF)

    We added performance breakdowns by flow magnitude and OD distance, together with qualitative examples, to show in which regimes GeoFlow improves performance.

- **Generalization / transferability** (Reviewer u6js)

    We complemented the original studies with geographic (longitudinal) partition experiments, demonstrating stable performance when transferring across different subregions.

- **Scalability** (Reviewer u6js)

    We conducted model capacity scaling experiments with respect to network width and depth, showing monotone and stable performance gains as the model size increases.

- **Efficiency** (Reviewer Tew9, hQRv)

    We added detailed training/inference time, FLOPs, and memory profiling, showing GeoFlow is more efficient in terms of time per case than TransFlower.

- **Interpretability** (Reviewer u6js, 3QvF, hQRv)

    We added more performance analysis, along with axial-global attention visualizations and a case study, which illustrates intuitive hub–periphery patterns.

The rebuttal and revision add targeted experiments and clarifications corresponding one-to-one to these concerns, aiming to resolve the issues raised by the reviewers directly.

**What we changed in the revision (all marked in blue):** we (1) reorganized and clarified the encoder and decoder subsection to remove ambiguity, (2) added new geographic transferability evaluation via longitudinal partition, (3) added generation performance profiles by flow magnitude and OD distance, (4) added scalability scaling experiments, (5) added empirical compute profiling vs. TransFlower (w.r.t. time/FLOPs/memory), and (6) added an interpretability case study visualizing axial–global attention. (7) Other minor changes were made to ensure consistent content and style.

---

### Meta-Review · Area_Chair_3XD7 · 2026-01-07

**Summary:**

This paper introduces GeoFlow, a unified framework for OD flow prediction and generation that combines geospatial feature augmentation, a geometric–intrinsic fusion encoder, and an axial–global attention decoder. Reviewers found the approach technically sound and empirically strong, but raised concerns regarding architectural clarity, computational complexity, fairness of baseline comparisons, interpretability, generalization, and practical efficiency. The authors convincingly resolved issues related to architectural correctness, global attention complexity, and baseline fairness through clearer exposition, additional ablations, and targeted comparisons. They also strengthened the paper with new performance analyses, scalability experiments, and qualitative attention visualizations.
Some concerns remain only partially addressed. While geographic transferability within commuting data is supported by additional partition experiments, generalization to other OD flow types, such as tourism or logistics, remains untested. Interpretability improvements are primarily qualitative, and efficiency comparisons are empirically demonstrated only against TransFlower, with broader baseline comparisons remaining indirect.

**Reviewer Concerns:**

Resolved Concerns

Architecture correctness and presentation (Reviewers 3QvF, Tew9)
Concerns regarding ambiguous notation, potential circular dependencies between encoder and decoder attention, and unclear definitions of intermediate tensors were addressed through clarifications and improved explanations. The revised description resolves the confusion around Equations 6–7 and the interaction between encoder-side and decoder-side axial attention.

Global attention complexity (Reviewer 3QvF)
The concern about potential O(N^4) complexity was resolved. The authors clarified that global attention operates on pooled area-level embeddings rather than all OD pairs, followed by broadcasting back to the OD grid. This explanation is coherent and supported by empirical profiling demonstrating practical feasibility.

Fairness of comparison with TransFlower (Reviewer 3QvF)
The concern that GeoFlow’s gains may stem primarily from added geometric features was addressed by introducing an augmented TransFlower baseline using the same features. The results show only marginal improvement for TransFlower, while GeoFlow retains a clear performance advantage, resolving this concern.

Performance analysis and failure regimes (Reviewers u6js, 3QvF)
Requests for deeper analysis of where performance gains occur were largely addressed. The authors added breakdowns by OD distance and flow magnitude and linked these to qualitative examples. The analysis clarifies that improvements concentrate on medium-to-high flow and typical-distance OD pairs, while limitations on sparse or extreme cases are transparently documented.

Geographic generalization within commuting data (Reviewer u6js)
The authors added geographic partition experiments that demonstrate stable performance across spatial splits within the commuting dataset. This adequately addresses concerns about spatial transferability across regions.

Partially or Unresolved Concerns

Cross-domain generalization beyond commuting OD data (Reviewer hQRv)
While the rebuttal strengthens evidence for spatial generalization within commuting flows, it does not provide empirical validation on fundamentally different OD domains such as tourism or logistics. The response relies on the lack of standardized benchmarks rather than experimental evidence. As a result, generalization across distinct OD flow types remains unverified and should be acknowledged as an open limitation.

Efficiency comparisons across broader baselines (Reviewers Tew9, hQRv)
Although the authors provide detailed profiling against TransFlower and show competitive wall-clock performance, efficiency comparisons are limited to a single baseline family. The authors note implementation differences as a constraint, but this leaves broader claims about efficiency relative to other OD models only partially supported.

Interpretability of attention mechanisms (Reviewers u6js, 3QvF, hQRv)
The added attention visualizations and qualitative case studies offer intuitive, domain-consistent insights (e.g., hub–periphery structure). However, interpretability remains illustrative rather than systematic, lacking causal or quantitative validation. As such, interpretability is improved but still limited in scope.

**Reviewer Scores:**

Reviewer u6js is likely to maintain a score of 6 after the discussion. Although questions and concerns are mostly addressed, a 6 is still reasonable.

Reviewer 3QvF is likely to raise the score from 4 to 6. The authors provided detailed clarifications of the encoder decoder structure, a clear explanation of global attention complexity, a fair baseline comparison with TransFlower augmented by the same features, and additional interpretability and performance analyses.

Review Tew9 is likely to maintain a score of 4 since the efficiency concerns remain partially solved.

Reviewer hQRv will also maintain a score of 4 because the cross-domain transferability remains unsolved.

---

### Decision · Program_Chairs · 2026-01-26

Reject